# Olica: Efficient Structured Pruning of Large Language Models without Retraining

Jiujun He [1]   Huazhen Lin [* 1]

## Abstract

Most existing structured pruning methods for Large Language Models (LLMs) require substantial computational and data resources for retraining to reestablish the corrupted correlations, making them prohibitively expensive. To address this, we propose a pruning framework for LLMs called **O**rthogonal decomposition and **li**near **ca**libration (Olica), which eliminates the need for retraining. A key observation is that the multi-head attention (MHA) layer depends on two types of matrix products (i.e., $\mathbf{W}_q \mathbf{W}_k^\top$ and $\mathbf{W}_v \mathbf{W}_o^\top$). By treating these matrix products as unified entities and applying principal component analysis (PCA), we extract the most important information to compress LLMs without sacrificing accuracy or disrupting their original structure. Consequently, retraining becomes unnecessary. A fast decomposition method is devised, reducing the complexity of PCA by a factor of the square of the number of attention heads. Additionally, to mitigate error accumulation problem caused by pruning the feed-forward network (FFN) layer, we introduce a linear calibration method to reconstruct the residual errors of pruned layerS using low-rank matrices. By leveraging singular value decomposition (SVD) on the solution of the least-squares problem, these matrices are obtained without requiring retraining. Extensive experiments show that the proposed Olica is efficient in terms of data usage, GPU memory, and running time, while delivering superior performance across multiple benchmarks.

---

[1]Center of Statistical Research, School of Statistics and Data Science, and New Cornerstone Science Laboratory, Southwestern University of Finance and Economics, Chengdu, China. Correspondence to: Huazhen Lin <linhz@swufe.edu.cn>.

*Proceedings of the $42^{st}$ International Conference on Machine Learning*, Vancouver, Canada. PMLR 267, 2025. Copyright 2025 by the author(s).

*Table 1.* We compare the resource consumption of different pruning methods on the LLaMA-7B model, focusing on the number of data usage, peak GPU memory consumption, and the runtime required for pruning (or retraining). The performance of the pruned model is evaluated based on perplexity (PPL) on the WikiText-2 dataset and accuracy averaged across the following datasets: BoolQ, PIQA, HellaSwag, WinoGrande, ARC-e, ARC-c, and OBQA. "SR" indicates the sparsity ratio of the pruned model.

| Method | Sam. | Time | Mem. | SR: 25% | | SR: 33% | |
|---|---|---|---|---|---|---|---|
| | | | | PPL ($\downarrow$) | Acc. ($\uparrow$) | PPL ($\downarrow$) | Acc. ($\uparrow$) |
| LLM-Pruner | 50K | 3h | 30GB | 20.57 | 58.67 | 24.50 | 55.39 |
| SlimGPT | 50K | 1h | 20GB | 18.45 | 62.45 | 22.43 | **61.41** |
| **Olica (Ours)** | **256** | **7min** | **3GB** | **16.69** | **63.53** | **19.83** | 61.21 |

## 1. Introduction

Since the introduction of transformer architecture (Vaswani et al., 2017), the field of natural language processing has witnessed a surge of unsupervised pre-training models, such as BERT (Devlin et al., 2019) and GPT series (Brown et al., 2020). Following the scaling law (Kaplan et al., 2020), the model parameters have expanded from hundreds of millions to hundreds of billions (Chowdhery et al., 2023; Touvron et al., 2023a; OpenAI, 2024), earning them the label of Large Language Models (LLMs). Despite emerging abilities such as in-context leaning and instruction following (Wei et al., 2022a) with increasing scale, the sheer size of LLMs makes their deployment and inference on edge devices highly challenging. To address this, techniques such as network pruning (Han et al., 2015; Frankle & Carbin, 2019), knowledge distillation (Hinton et al., 2015; Sun et al., 2019), and quantization (Frantar et al., 2023; Dettmers et al., 2024) have been proposed.

In this work, we focus mainly on network pruning, which aims to eliminate redundant parameters in neural networks while preserving their performance. Before the era of LLMs, network pruning approaches primarily relied on Taylor expansion of the loss function or regularization techniques. For instance, Optimal Brain Damage (LeCun et al., 1989), Optimal Brain Surgeon (Hassibi & Stork, 1992), and later works (Liu et al., 2021; Kwon et al., 2022) assess the the

importance of parameters using the Hessian matrix of the objective function. Despite their success, it is prohibitively expensive to acquire the gradient of full model parameters for LLMs. On the other hand, regularization-based approaches (Han et al., 2015; Wen et al., 2016; Tan et al., 2024) impose $l_1$ or $l_2$ penalties on the parameters of neural networks, thereby pruning those with relatively smaller magnitudes. However, the pre-training process of LLMs is so resource-intensive that developers often avoid imposing structured penalties. Penalty-based pruning methods have become less effective in the context of LLMs, as evidenced by empirical results shown in (Frantar & Alistarh, 2023).

Recent pruning works on LLMs can be categorized into unstructured and structured approaches. Unstructured pruning (Frantar & Alistarh, 2023; Sun et al., 2024) removes individual weights but often struggles to achieve significant speedup without specialized libraries or hardware. In contrast, structured pruning (Ma et al., 2023; Li et al., 2024; Zhao et al., 2024a; Ling et al., 2024; Gao et al., 2024) regularly reduces the dimensionality of intermediate features, allowing it to benefit from most GPU devices. However, existing structured pruning methods typically require substantial computational resources and large amounts of data for retraining to reestablish the corrupted correlations. For instance, DISP-LLM (Gao et al., 2024) necessitates 4 NVIDIA A100 80GB GPUs to prune a 13B LLaMA model, while methods like LLM-Pruner (Ma et al., 2023), LoRAP (Li et al., 2024), and SlimGPT (Ling et al., 2024) demand tens of thousands of well-annotated instruction data (i.e., Alpaca (Taori et al., 2023)) for retraining to recover the model's performance. These significant computational and data requirements present huge challenges in resource-constrained settings. For example, when pruning LLMs for a specific domain, annotating a large amount of instruction data from scratch is a prohibitively labor-intensive and time-consuming process.

To address these challenges, we propose an efficient structured pruning framework for LLMs that eliminates the need for retraining, called Orthogonal Decomposition and Linear Calibration (Olica). The key observation motivating our method is that the core design of transformers, i.e., the multi-head attention (MHA) layer, involves two types of matrix products (i.e., $\mathbf{W}_q\mathbf{W}_k^\top$ and $\mathbf{W}_v\mathbf{W}_o^\top$). We treat these products as unified entities and apply Principal Component Analysis (PCA) to extract the most important information for compressing LLMs. The benefit of this approach is that it eliminates the data requirements for reestablishing correlations within the MHA layer, as it directly operates on the parameter matrices. Furthermore, to reduce the complexity of PCA, we devise a fast-approach that only performs SVD on one of the product matrices based on the observation of similar distributions of singular values for $\mathbf{W}_q$ and $\mathbf{W}_k$ (also for $\mathbf{W}_v$ and $\mathbf{W}_o$). The fast-approach reduces

the complexity from $O(h \cdot d^3)$ to $O(d^3/h)$, where $d$ and $h$ are the dimensionality of the embedding and the number of heads, respectively. In addition, pruning a layer can alter its output, and these changes may be amplified as layers stack. To mitigate error accumulation problem caused by pruning, we introduce a linear calibration strategy to reconstruct the residual errors of the pruned feed-forward network (FFN) layers. Our approach leverages the closed-form solution of ridge regression to model the reconstruction process, thereby alleviating the need for retraining. Additionally, to reduce the number of additional parameters introduced by the linear calibration, we apply a low-rank approximation to the ridge regression solution.

To summarize, our contributions are as follows: **i)** We propose applying PCA to the matrix product in the MHA layer, which compresses the model without the need for retraining. Moreover, we devise a fast decomposition method, reducing the complexity of PCA by a factor of the square of the number of attention heads. **ii)** We introduce linear calibration to model the residual errors of pruned FFN layers. Along with the proposed weighted SVD and layer selection criterion, this method adds negligible extra parameters. **iii)** Our approach is efficient in terms of data usage, GPU memory consumption, and runing time, while achieves performance that is either better or on par with existing methods across several benchmarks. For example, as shown in Table 1, compared to state-of-the-art structured pruning methods (e.g., LLM-Pruner (Ma et al., 2023) and SlimGPT (Ling et al., 2024)), Olica requires only hundreds of calibration samples, consumes less GPU memory, and significantly reduces the running time while delivering better performance.[1]

## 2. Related Work

**Large Language Models (LLMs).** As demonstrated in (Zhao et al., 2024b), the development of Language Models (LMs) follows four major stages: statistical-based, neural-based, pre-training-based, and scaling-based LMs. The rise of LLMs is primarily concentrated in the third stage. Pre-training-based LMs, such as BERT (Devlin et al., 2019; Liu, 2019), GPT-1 (Radford et al., 2018), and GPT-2 (Radford et al., 2019), typically follow a paradigm where the model is first pre-trained on a large corpus in a task-agnostic manner, and then fine-tuned for specific downstream tasks. Subsequently, according to the scaling law (Kaplan et al., 2020), researchers have found that by simply scaling up the size of the pre-trained model and the amount of pre-training data, and with sufficient computational resources, the performance of LMs increases exponentially (Wei et al., 2022a; Schaeffer et al., 2023). Through scaling, LMs have demonstrated exceptional abilities in handling downstream tasks,

---

[1]Code is available at `https://github.com/BetterTMrR/LLM-Olica`.

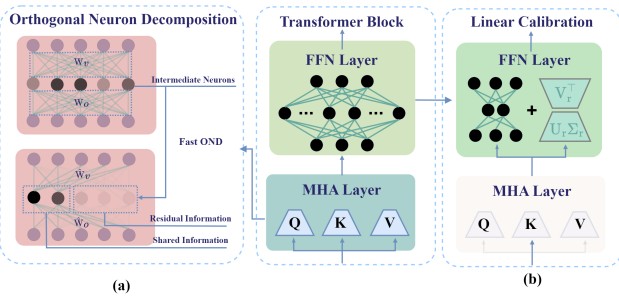

Figure 1. (**a**): Orthogonal neuron decomposition. Neurons that carry shared information preserve the core projection information from all the original neurons, while neurons that carry residual information can be pruned with minimal impact on accuracy. (**b**): Linear calibration. This is an example of reconstructing the residual errors of pruned FFN Layers.

such as in-context learning (Brown et al., 2020), instruction following (Ouyang et al., 2022), and chain-of-thought prompting (Wei et al., 2022b). As a result, the parameters of current LLMs generally range from tens of billions to hundreds of billions (Touvron et al., 2023a; Bai et al., 2023; OpenAI, 2024; Team, 2024).

Despite LLMs surpassing human performance across various tasks, their large parameter scale makes deployment and inference highly challenging, hindering researchers from conducting further studies on these models.

**Network Pruning on LLMs.** Network pruning aims to remove redundant parameters from neural networks while maximally maintaining their performance. Pioneering pruning works (Ma et al., 2023; Frantar & Alistarh, 2023) have identified two major challenges in pruning LLMs: the overwhelmingly large number of parameters and the lack of readily available training data. These challenges make conventional gradient-based (LeCun et al., 1989; Hassibi & Stork, 1992; Molchanov et al., 2019; Sanh et al., 2020) and penalty-based (Han et al., 2015; Wen et al., 2016; Tan et al., 2025) methods inefficient in the context of LLMs. To address these issues, Ma et al. (2023) proposed a two-stage pruning framework: first, using a small amount of calibration data to eliminate unimportant components of LLMs (fast), and then leveraging Low-rank Adaptation (LoRA) (Hu et al., 2022) to retrain the pruned model and recover its performance (time-consuming). Most subsequent works on structured pruning (Ashkboos et al., 2024; Ling et al., 2024; Li et al., 2024) have followed this paradigm, with differences in the first phase. For instance, SliceGPT (Ashkboos et al., 2024) uses PCA to reduce the dimensionality of the hidden representations; LoRAP (Li et al., 2024) employs low-rank approximation to replace the weight matrices in the self-attention layer; and SlimGPT (Ling et al., 2024) extends the optimal brain surgeon (OBS) framework (Has-

sibi & Stork, 1992; Frantar & Alistarh, 2023) to structured pruning scenarios. Additionally, Compresso (Guo et al., 2023), LoRAPrune (Zhang et al., 2024), and APT (Zhao et al., 2024a) proposed single-stage pruning approaches, but these still require LoRA to retrain the LLMs.

As mentioned in Section 1, retraining LLMs using LoRA is a tedious and resource-intensive process, especially for models with tens of billions of parameters. In contrast, our proposed approach has the distinct feature of single-stage pruning without the need for retraining.

## 3. Methodology

### 3.1. Preliminaries

**Transformer.** The transformer model consists of $L$ blocks, each of which contains two consecutive layers: a multi-head self-attention (MHA) layer and a feed-forward network (FFN) layer. Let $\mathbf{X}_l \in \mathbb{R}^{n \times d}$ represent the inputs to the $l^{\text{th}}$ transformer layer, where $n$ is the length of the token sequence and $d$ is the dimensionality of a token. The forward pass of the $l^{\text{th}}$ transformer block is then given by:

$$\begin{aligned} \mathbf{X}^{\text{MHA}} &= \mathbf{X}_l + \text{MHA}(\mathbf{X}_l), \\ \mathbf{X}_{l+1} &= \mathbf{X}^{\text{MHA}} + \text{FFN}(\mathbf{X}^{\text{MHA}}), \end{aligned} \quad (1)$$

where $\text{MHA}(\cdot)$ and $\text{FFN}(\cdot)$ are defined as:

$$\text{MHA}(\mathbf{X}) = \sum_{i=1}^{h} \text{SM}\left(\frac{\mathbf{X}\mathbf{W}_{q_i}\mathbf{W}_{k_i}^{\top}\mathbf{X}^{\top}}{\sqrt{d_h}}\right)\mathbf{X}\mathbf{W}_{v_i}\mathbf{W}_{o_i}^{\top}, \quad (2)$$

$$\text{FFN}(\mathbf{X}) = \sigma(\mathbf{X}\mathbf{W}_u)\mathbf{W}_d^{\top}, \quad (3)$$

where SM is the softmax operation. $\mathbf{W}_{q_i}$, $\mathbf{W}_{k_i}$, $\mathbf{W}_{v_i}$, $\mathbf{W}_{o_i} \in \mathbb{R}^{d \times d_h}$ denote the $i^{th}$ head's query, key, value, and output matrices, respectively. $h$ and $d_h$ are the number of heads and the dimensionality of each head (generally $d = d_h \times h$), correspondingly. $\mathbf{W}_u, \mathbf{W}_d \in \mathbb{R}^{d \times 4d}$ and $\sigma$ is an activation function, such as ReLU, GeLU, SiLU, etc. Some LLMs, like LLaMA, adopt a gated activation: $\text{FFN}(\mathbf{X}) = (\mathbf{X}\mathbf{W}_u \odot \sigma(\mathbf{X}\mathbf{W}_g))\mathbf{W}_d^{\top}$, where $\odot$ denotes the Hadamard product.

**Importance Scores.** In network pruning works, the most common criterion for evaluating the importance of elements in the weight matrix $\mathbf{W}$ is the first-order Taylor expansion of the loss function $\mathcal{L}(\cdot)$: $|\mathcal{L}(\theta|\mathbf{W}_{ij} = 0) - \mathcal{L}(\theta)| \approx |\mathbf{W}_{ij}\frac{\partial\mathcal{L}(\theta)}{\partial\mathbf{W}_{ij}}|$, where $\theta$ is the parameter set that consists of all weight matrices. This criterion reflects how much the loss function changes when setting $\mathbf{W}_{ij} = 0$.

The above criterion requires computing the gradient of all parameters in the model, which is computationally expensive for LLMs. To mitigate this, Wanda (Sun et al., 2024)

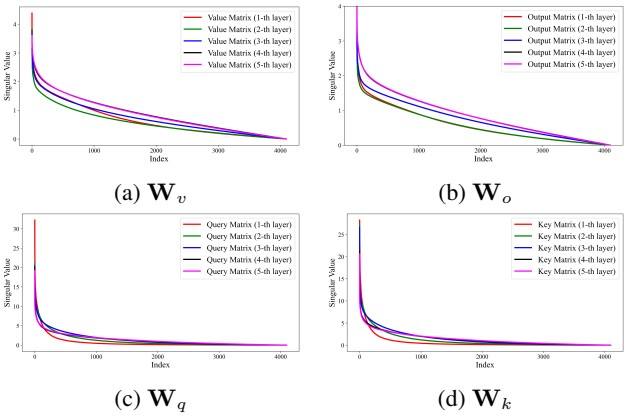

(a) $\mathbf{W}_v$                    (b) $\mathbf{W}_o$

(c) $\mathbf{W}_q$                    (d) $\mathbf{W}_k$

*Figure 2.* The distributions of singular values of different weight matrix in the MHA layer (LLaMA-7B). $\mathbf{W}_q$ and $\mathbf{W}_k$ exhibit strongly similar distribution of singular values, and similar results can be observed from $\mathbf{W}_v$ and $\mathbf{W}_o$. Notably, $\mathbf{W}_q$ and $\mathbf{W}_k$ show more significant low-rank property.

proposed using the magnitudes of weights and activations to measure the contribution of a parameter $\mathbf{W}_{ij}$:

$$\mathcal{I}(\mathbf{W}_{ij}) = \|\boldsymbol{x}^{(i)}\|_2 \cdot |\mathbf{W}_{ij}|, \qquad (4)$$

where $\|\boldsymbol{x}^{(i)}\|_2$ is the $l_2$ norm of the $i^{th}$ column of $\mathbf{X}$, i.e., the vector comprised of $i^{th}$ feature of all $n$ tokens. Formula (4) is intuitive: if both the $i^{\text{th}}$ feature of the input samples and $\mathbf{W}_{ij}$ have larger magnitudes, removing the parameter $\mathbf{W}_{ij}$ will result in a drastic change in the projection output $\mathbf{XW}$, so it should be kept. Formula (4) is fast and memory-efficient for assessing the importance of parameters in LLMs, as it only requires the forward pass of the model. Therefore, in this work, we use (4) by default to evaluate the importance scores of the model's parameters. Although originally designed for unstructured pruning, the subsequent work LoRAP (Li et al., 2024) has demonstrated that it is also effective for structured pruning. Inspired by these empirical results, we measure the importance of a group of parameters, such as the $j^{\text{th}}$ intermediate neuron of the FFN layer, as follows:

$$\mathcal{I}(\text{neuron}_j) = \sum_i [\mathcal{I}(\mathbf{W}_{u_{ij}}) + \mathcal{I}(\mathbf{W}_{d_{ij}})]. \qquad (5)$$

### 3.2. Orthogonal Decomposition for MHA

In this section, we consider the compression of the MHA layer. From (2), we can see that the MHA layer, the key design of transformers, depends on matrices $\mathbf{W}_q$, $\mathbf{W}_k$, $\mathbf{W}_v$ and $\mathbf{W}_o$ through two kinds of matrix product: $\mathbf{W}_{qk} = \mathbf{W}_q \mathbf{W}_k^\top$ and $\mathbf{W}_{vo} = \mathbf{W}_v \mathbf{W}_o^\top$, where the head index is omitted for simplicity. Thus, $\mathbf{W}_{qk}$ and $\mathbf{W}_{vo}$ can be treated as unified entities. The most important information about a specific MHA layer then can be derived by analyzing $\mathbf{W}_{qk}$

and $\mathbf{W}_{vo}$. Here, we use PCA to extract the most important features from $\mathbf{W}_{qk}$ and $\mathbf{W}_{vo}$ to compress the models while maintaining the performance of the MHA layer. Specifically, taking $\mathbf{W}_{vo}$ as an example, we apply SVD to $\mathbf{W}_{vo}$: $\mathbf{W}_{vo} = \mathbf{U}\boldsymbol{\Sigma}\mathbf{V}^\top$, and define $\hat{\mathbf{W}}_v \leftarrow \mathbf{U}\boldsymbol{\Sigma}$ and $\hat{\mathbf{W}}_o \leftarrow \mathbf{V}$. Clearly, we have $\hat{\mathbf{W}}_v \hat{\mathbf{W}}_o^\top = \mathbf{W}_v \mathbf{W}_o^\top$. Since the columns of $\mathbf{U}$ and $\mathbf{V}$ are orthogonal, the resulting output neurons, such as $\mathbf{X}\hat{\mathbf{W}}_v$, carry distinct information, as shown in Figure 1 **(a)**. This ensures that the information is captured as fully as possible within the given dimensionality. We refer to this method as Orthogonal Neuron Decomposition (OND). One of the advantages of OND is that it can extract the key information from the MHA layer without requiring additional data and retraining.

**MHA Pruning.** A direct approach for pruning the MHA layer is to preserve the first $r$ eigenvectors corresponding to the largest eigenvalues when performing SVD. This is known as the magnitude-based pruning method. However, eigenvalues with smaller magnitudes may still contain important information. Therefore, we use (5) to evaluate the importance of each eigenvector, resulting in the following criteria:

$$\mathcal{I}(\text{neuron}_j) = \sum_i [\mathcal{I}(\hat{\mathbf{W}}_{v_{ij}}) + \mathcal{I}(\hat{\mathbf{W}}_{o_{ij}})]. \qquad (6)$$

Based on (6), we prune the neurons with the least importance, thereby reducing the dimensionality of each attention head while maintaining performance.

**Fast OND.** The complexity of performing SVD on $\mathbf{W}_{vo} \in \mathbb{R}^{d \times d}$ is $O(d^3)$. Moreover, there are $h$ heads in a MHA layer, resulting in a complexity of $O(hd^3)$. This complexity requires about an hour to prune a 7B model. Fortunately, we observe similar distributions of singular values for $\mathbf{W}_v$ and $\mathbf{W}_o$ (also for $\mathbf{W}_q$ and $\mathbf{W}_k$) as shown in Figure 2. This means that the number of retained singular values required to preserve a certain energy ratio for $\mathbf{W}_v$ and $\mathbf{W}_o$ is also similar. Therefore, we can only perform SVD on one of $\mathbf{W}_v$ and $\mathbf{W}_o$ to roughly determine the redundancy of the unified entity $\mathbf{W}_{vo}$. Specifically, for instance, letting $\mathbf{W}_v = \mathbf{U}\boldsymbol{\Sigma}\mathbf{V}^\top$, then we have $\hat{\mathbf{W}}_v \leftarrow \mathbf{U}$ and $\hat{\mathbf{W}}_o \leftarrow \mathbf{W}_o \mathbf{V}\boldsymbol{\Sigma}^\top$. Since typically $\mathbf{W}_v \in \mathbb{R}^{d \times d/h}$, the SVD complexity is $O(d^3/h^2)$, leading to a complexity of $O(d^3/h)$ for a entire MHA layer. Fast-OND reduces the complexity by a factor of $h^2$. In the LLaMA-7B model, the number of heads $h$ is 32, and pruning it requires only several minutes of runtime.

### 3.3. Pruning and Linear Calibration for FFN

**FFN Pruning.** As illustrated in Figure 1 **(b)**, our goal is to prune the intermediate neurons of each FFN layer, making the pruned FFN layer slimmer. To achieve this, we evaluate their importance scores using (5) and prune the neurons with the lowest importance scores. For example, with a specified

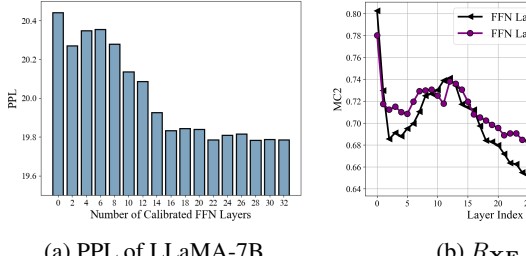

(a) PPL of LLaMA-7B      (b) $R_{\mathbf{XE}}$

*Figure 3.* (a): Perplexity (PPL, lower is better) on WikiText2 with different number of FFN layers calibrated. (b): Each FFN layer's MC2 of LLaMA-7B and LLaMA-13B models.

sparsity ratio $s$, the weights of the pruned FFN layer can be represented as $\hat{\mathbf{W}}_u, \hat{\mathbf{W}}_d \in \mathbb{R}^{d \times \hat{d}}$, where $\hat{d} = \lfloor (1-s) \times 4d \rfloor$, and the original weights are $\mathbf{W}_u, \mathbf{W}_d \in \mathbb{R}^{d \times 4d}$.

**Linear Calibration**. Pruning a layer generally leads to significant changes in its output. Conventional methods typically require retraining to restore the correlation between layers. Here, we propose a linear calibration strategy to reconstruct the residual errors (RE) of the pruned FFN layers, eliminating the need for retraining. Let $f$ denote a FFN layer, and let $\hat{f}$ represent the pruned version of $f$. To mitigate these changes, we use a linear model to recover the alterations by solving the ridge regression loss:

$$\hat{\mathbf{W}} = \arg\min_{\mathbf{W} \in \mathbb{R}^{d \times d}} \|\mathbf{E} - \mathbf{X}\mathbf{W}\|_2^2 + \lambda \|\mathbf{W}\|_F^2, \quad (7)$$

where $\mathbf{E} = f(\mathbf{X}) - \hat{f}(\mathbf{X}) \in \mathbb{R}^{n \times d}$ indicates the RE of a pruned layer $\hat{f}$, $\|\cdot\|_F$ is the Frobenius norm, and $\lambda$ indicates the penalty strength, which helps avoid the irreversible problem. Then, the forward pass of our linear calibration for the $l^{th}$ layer is as follows:

$$\mathbf{X}_{l+1} = \hat{f}_l(\mathbf{X}_l) + \mathbf{X}_l \hat{\mathbf{W}}_l. \quad (8)$$

The loss of ridge regression (7) has a closed-form solution $\hat{\mathbf{W}} = (\mathbf{X}^\top \mathbf{X} + \lambda \mathbf{I})^{-1} \mathbf{X}\mathbf{E}$, where $\mathbf{I}$ is an identity matrix. Therefore, we can use a small amount of calibration data to obtain the solution, avoiding the need for retraining.

**Layer Selection.** Figure 3 (a) shows that perplexity (PPL, lower is better) decreases, then stays stable as the number of calibrated FFN layers increases. This implies that more calibration is not necessarily better, especially it may introduce more parameters. Hence, it is necessary to select the FFN layers that should be calibrated. Here, we choose layers whose RE are linearly recoverable, as we are using the linear calibration strategy. To this end, we propose a criterion based on the Multiple Correlation Coefficient (MC2), which is used to express the linear correlation and is denoted as $R_{\mathbf{XE}}$. Let $\hat{\mathbf{E}} = \mathbf{X}\hat{\mathbf{W}} \in \mathbb{R}^{n \times d}$ represents the prediction of a linear model that fits the residual data $(\mathbf{X}, \mathbf{E})$. Then, the

$R_{\mathbf{XE}}$ is defined as follows:

$$R_{\mathbf{XE}} = \frac{1}{d} \sum_{i=1}^{d} R_i \quad (9)$$

where $R_i$ is the pearson correlation coefficient:

$$R_i = \frac{\sum_{j=1}^{n}(\boldsymbol{e}_j^{(i)} - \bar{\boldsymbol{e}}^{(i)})(\hat{\boldsymbol{e}}_j^{(i)} - \bar{\hat{\boldsymbol{e}}}^{(i)})}{\sqrt{\sum_{j=1}^{n}(\boldsymbol{e}_j^{(i)} - \bar{\boldsymbol{e}}^{(i)})^2 \sum_{j=1}^{n}(\hat{\boldsymbol{e}}_j^{(i)} - \bar{\hat{\boldsymbol{e}}}^{(i)})^2}}, \quad (10)$$

$\boldsymbol{e}^{(i)}$ is the $i^{th}$ column of $\mathbf{E}$, $\bar{\boldsymbol{e}}^{(i)}$ indicates its mean, and similarly for $\hat{\boldsymbol{e}}^{(i)}$ and $\bar{\hat{\boldsymbol{e}}}^{(i)}$ when $\mathbf{E}$ is replaced by $\hat{\mathbf{E}}$. If $R_i$ is large, a linear model can effectively approximate $\boldsymbol{e}^{(i)}$: $\boldsymbol{e}^{(i)} \approx \mathbf{X}\boldsymbol{w}^{(i)}$ where $\boldsymbol{w}^{(i)}$ indicates the $i^{th}$ column of $\hat{\mathbf{W}}$. Therefore, a large value of (9) indicates that the RE of the pruned layer $\hat{f}$ is able to be calibrated. Figure 3 (b) shows the $R_{\mathbf{XE}}$ values for each FFN layer of the LLaMA-7B and LLaMA-13B models, demonstrating that the RE of the FFN layer in the shallower blocks are highly calibratable.

**Low-rank Approximation.** The linear calibration introduces a matrix $\hat{\mathbf{W}} \in \mathbb{R}^{d \times d}$, with the number of parameters being $d^2$. This accounts for 1/8 of the number of parameters for a FFN layer. To reduce the number of extra parameters introduced, we propose to use a low-rank approximation of $\hat{\mathbf{W}}$. Specifically, let $\hat{\mathbf{W}} = \mathbf{U}\boldsymbol{\Sigma}\mathbf{V}^\top$, and define $\hat{\mathbf{W}}_1 = \mathbf{U}_r\boldsymbol{\Sigma}_r$ and $\hat{\mathbf{W}}_2 = \mathbf{V}_r$, where we preserve the first $r$ eigenvectors with the largest eigenvalues. Then, the number of parameters, i.e., $\hat{\mathbf{W}}_1$ and $\hat{\mathbf{W}}_2$, introduced by the linear calibration is $2dr$, which is far less than $d^2$ if we take $r \ll d$. The forward pass of (8) can then be modified as:

$$\mathbf{X}_{l+1} = \hat{f}_l(\mathbf{X}_l) + \mathbf{X}_l \hat{\mathbf{W}}_{l_1} \hat{\mathbf{W}}_{l_2}^\top. \quad (11)$$

## 4. Experiments

### 4.1. Experimental Settings

**Models and evaluation protocols**. Following the previous works (Ma et al., 2023; Li et al., 2024), we mainly focus on the evaluation of LLaMA-1 (Touvron et al., 2023a), LLaMA-2 (Touvron et al., 2023b), and Vicuna (Chiang et al., 2023) models. We assess the performance of the pruned models on WikiText2 (Merity et al., 2017) with sequence length of 128 tokens, and on the following downstream tasks: BoolQ (Clark et al., 2019), PIQA (Bisk et al., 2020), HellaSwag (Zellers et al., 2019), WinoGrande (Sakaguchi et al., 2021) , ARC-easy (Clark et al., 2018), ARC-challenge (Clark et al., 2018), and OpenbookQA (Mihaylov et al., 2018). We employ lm-eval-harness framework (Gao et al., 2021) to evaluate the pruned model performance on these tasks[2].

---

[2]The version of lm-eval-harness used in this paper is the same as SlimGPT (Ling et al., 2024), which can be found in their the sup-

*Table 2.* Zero-shot performance and PPL on WikiText2 of pruned LLaMA-1 family. "SR" indicates sparsity ratio, and "FT" denotes whether LoRA fine-tuning is used to recover the model's performance. † represents results that are reproduced by this paper. The best and the second best performances are marked as **bolded** and underlined, correspondingly.

| Model | SR% | Method | FT | PPL (↓) | BoolQ | PIQA | HellaS | WinoG | ARC-e | ARC-c | OBQA | Avg. (↑) |
|---|---|---|---|---|---|---|---|---|---|---|---|---|
| LLaMA-7B | 0% | Dense | ✗ | 12.63 | 75.08 | 79.16 | 76.20 | 70.00 | 72.89 | 44.88 | 44.40 | 66.09 |
| | 20% | LLM-Pruner | ✓ | 18.01 | 66.76 | 78.45 | 71.44 | 63.77 | 66.41 | 39.85 | 43.80 | 61.50 |
| | | DISP-LLM | ✗ | - | - | 76.66 | 68.39 | 64.72 | 64.81 | 37.12 | - | - |
| | | Compresso | ✓ | - | **79.08** | 75.46 | 53.44 | 67.80 | 68.64 | 37.97 | 34.20 | 59.51 |
| | | FLAP | ✗ | 17.00 | 69.40 | 74.70 | 66.90 | 66.30 | 64.60 | 36.50 | 38.20 | 59.50 |
| | | LoraPrune | ✓ | 16.80 | 65.62 | **79.31** | 70.00 | 62.76 | 65.87 | 37.69 | 39.14 | 60.06 |
| | | LoRAP † | ✗ | 15.84 | 73.88 | 76.71 | 72.94 | 68.03 | 69.70 | 40.87 | 42.40 | 63.60 |
| | | SlimGPT | ✗ | 16.99 | 75.93 | 77.58 | 73.07 | 67.96 | 68.60 | 41.72 | 41.80 | 63.81 |
| | | Olica (Ours) | ✗ | **15.35** | 71.59 | 77.91 | **73.30** | **70.01** | **72.10** | **42.66** | **44.20** | **64.54** |
| | 25% | LLM-Pruner | ✓ | 20.57 | 62.81 | 76.93 | 69.21 | 60.46 | 63.34 | 38.14 | 39.80 | 58.67 |
| | | Compresso | ✓ | - | 73.55 | 73.07 | 49.16 | 64.80 | 66.20 | 37.20 | 29.80 | 56.25 |
| | | SlimGPT | ✗ | 19.11 | **75.11** | 76.77 | 70.60 | 67.25 | 66.75 | 40.40 | 40.40 | 62.47 |
| | | LoRAP † | ✗ | 17.40 | 72.14 | **77.20** | 71.30 | **68.75** | 68.48 | 39.16 | 41.00 | 62.57 |
| | | Olica (Ours) | ✗ | **16.69** | 72.54 | 76.88 | **71.40** | 68.19 | **70.83** | **41.72** | **43.20** | **63.54** |
| | 33% | LLM-Pruner | ✓ | 24.50 | 62.02 | 74.92 | 64.41 | 61.80 | 53.79 | 32.00 | 38.80 | 55.39 |
| | | Compresso | ✓ | - | 68.69 | 72.85 | 47.18 | 63.38 | 65.99 | 35.07 | 29.00 | 54.59 |
| | | LoRAP † | ✗ | 21.66 | 66.54 | 74.70 | 67.11 | 65.98 | 63.97 | 35.41 | 39.40 | 59.02 |
| | | SlimGPT | ✗ | 24.55 | 72.72 | **75.68** | **68.10** | 66.54 | 62.29 | 37.03 | 40.20 | 60.37 |
| | | Olica (Ours) | ✗ | **19.83** | **72.87** | 75.55 | 67.95 | **67.01** | **66.25** | **37.63** | **41.20** | **61.21** |
| LLaMA-13B | 0% | Dense | ✗ | 11.58 | 77.89 | 80.14 | 79.06 | 72.85 | 74.75 | 47.61 | 44.80 | 68.16 |
| | 20% | LLM-Pruner | ✓ | 16.62 | **79.38** | 77.36 | 71.47 | 70.32 | 70.54 | 44.88 | **45.80** | 65.68 |
| | | SlimGPT | ✗ | 14.87 | 77.06 | **79.82** | 76.94 | 72.61 | 69.78 | 44.80 | 43.60 | 66.37 |
| | | LoRAP † | ✗ | 13.84 | 78.87 | 79.05 | **77.54** | 71.35 | 73.57 | 43.00 | 44.54 | 66.84 |
| | | Olica (Ours) | ✗ | **13.68** | 78.32 | 78.89 | 77.21 | **74.11** | **73.99** | **46.59** | 44.60 | **67.67** |

**Implementation details.** We randomly select 256 samples from Bookcorpus (Zhu et al., 2015) and Alpaca (Taori et al., 2023) datasets, each of which is truncated to a sequence length of 128 tokens, as the calibration data. The number of calibrated FFN layers is selected from {6, 12, 16} for models with different parameter sizes. In the linear calibration, we retain the top 3% eigenvectors for low-rank approximation, i.e., $r/d = 0.03$. Following LoRAP (Li et al., 2024), we compress the MHA and FFN layers, until achieving a specified sparsity ratio of the entire model (i.e., including the token embedding layer and the final projection layer). All the experiments are conducted on a single NVIDIA A100 80GB GPU. More details can be found in Appendix A, and a detailed algorithm can be found in Appendix B.

plementary material: `https://openreview.net/forum?id=MxF0IKJtKW`. Therefore, the baseline results in this paper, if not specified, are cited from (Ling et al., 2024).

**Baselines.** We select state-of-the-art structured pruning approaches for comparisons: LLM-Pruner (NeuIPS'23) (Ma et al., 2023), LoraPrune (Findings of ACL'23) (Zhang et al., 2024), Compresso (Guo et al., 2023), FALP (AAAI' 24) (An et al., 2024), SliceGPT (ICLR'24) (Ashkboos et al., 2024), LLM-Surgeon (ICLR'24) (van der Ouderaa et al., 2024), LoRAP (ICML'24) (Li et al., 2024), DISP-LLM (NeuIPS'24) (Gao et al., 2024), and SlimGPT (NeuIPS'24) (Ling et al., 2024).

### 4.2. Main Results

**Performance.** The experimental results presented in Table 2 and Table 3 highlight the superior performance of our Olica method across multiple models and sparsity ratios. In particular, for LLaMA-7B in Table 2, Olica achieves the highest average accuracy of 64.54%, 63.54%, and 61.21%, while achieving the lowest perplexity (PPL) values of 15.35,

*Table 3.* Zero-shot performance and PPL on WikiText2 of pruned LLaMA-2 and Vicuna. "SR" indicates sparsity ratio, and "FT" denotes whether LoRA fine-tuning is used to recover the model's performance. † represents results that are reproduced by this paper. The best and the second best performances are marked as **bolded** and underlined, correspondingly.

| Model | SR% | Method | FT | PPL (↓) | BoolQ | PIQA | HellaS | WinoG | ARC-e | ARC-c | OBQA | Avg. (↑) |
|---|---|---|---|---|---|---|---|---|---|---|---|---|
| LLaMA-2-7B | 0% | Dense | ✗ | 12.19 | 77.71 | 79.05 | 76.00 | 68.98 | 74.58 | 46.33 | 44.20 | 66.69 |
| | 30% | SliceGPT | ✗ | - | - | 63.55 | 49.62 | 61.33 | 51.77 | 31.23 | - | - |
| | | DISP-LLM | ✗ | - | - | 73.72 | 62.87 | 63.93 | 60.10 | 37.03 | - | - |
| | | LLM-Surgeon | ✗ | - | 61.25 | 73.56 | 60.72 | 61.09 | 63.09 | 36.69 | 38.80 | 56.56 |
| | | LoRAP † | ✗ | 19.42 | 68.93 | **75.46** | 66.66 | 66.30 | 62.37 | 35.49 | 37.00 | 58.89 |
| | | Olica (Ours) | ✗ | **18.54** | **71.19** | 75.35 | **67.04** | **67.88** | **68.60** | **37.71** | **39.20** | **61.14** |
| Vicuna-7B | 0% | Dense | ✗ | 16.11 | 78.41 | 78.56 | 74.68 | 70.09 | 72.01 | 43.77 | 43.40 | 65.85 |
| | 20% | LLM-Pruner | ✓ | 19.11 | 61.96 | 76.88 | 69.18 | 63.30 | 61.83 | 37.88 | 39.40 | 58.63 |
| | | SlimGPT | ✗ | 21.14 | 75.41 | **77.09** | 72.34 | 68.43 | 69.23 | 41.47 | 43.40 | 63.91 |
| | | LoRAP † | ✗ | 20.74 | **78.01** | 76.12 | 72.62 | 66.93 | 71.17 | **43.09** | 42.80 | 64.39 |
| | | Olica (Ours) | ✗ | **20.23** | 75.81 | 76.66 | **72.82** | **68.67** | **72.81** | 42.41 | **45.00** | **64.88** |

*Table 4.* Statistics of the pruned model LLaMA-7B, including the parameters, MACs, memory, and inference latency.

| SR% | Params | MACs | Memory | Latency |
|---|---|---|---|---|
| 0% | 6.74B | 424.02G | 12884.5MiB | 46.95s |
| 20% | 5.39B | 373.23G | 10464.3MiB | 40.62s |
| 25% | 5.01B | 360.25G | 9696.3MiB | 39.32s |
| 33% | 4.52B | 339.53G | 8718.1MiB | 35.78s |

*Table 5.* Compare Fast-OND with SVD, Activation-Weighted SVD (AWSVD) (Li et al., 2024), and Wanda (Sun et al., 2024), under scenarios with and without linear calibration (LC).

| Setting | Method | PPL (↓) | Mean Accuracy (↑) |
|---|---|---|---|
| w/o LC | SVD | 71.01 | 47.62 |
| | AWSVD | 25.78 | 59.93 |
| | Wanda | 20.94 | 59.82 |
| | Fast-OND | **20.34** | **60.68** |
| w/ LC | SVD | 63.48 | 48.62 |
| | AWSVD | 24.72 | 60.38 |
| | Wanda | 20.40 | 60.04 |
| | Fast-OND | **19.83** | **61.21** |

16.69, and 19.83 at sparsities of 20%, 25%, and 33%, respectively. Additionally, the improvement of Olica over other methods increases with higher sparsity. These demonstrate its ability to maintain both modeling and reasoning performance under compression and pruning. For LLaMA-13B at 20% sparsity, Olica outperforms all baseline methods with the highest average accuracy of 67.67% and the lowest PPL of 13.68, demonstrating its scalability to larger models. In Table 3, for LLaMA-2-7B at 30% sparsity, Olica achieves the best average accuracy of 61.14% and the lowest PPL of 18.54, outperforming structured pruning methods like LLM-Surgeon and LoRAP across key benchmarks. Similarly, for Vicuna-7B at 20% sparsity, Olica leads with an average accuracy of 64.88%, excelling on datasets like WinoG (68.67%) and OBQA (45.00%), while maintaining the lowest PPL of 20.23. These results demonstrate that Olica consistently outperforms baseline pruning methods across diverse tasks, higher sparsity ratios, and multiple model architectures, all while requiring no retraining to recover performance. This makes Olica a highly effective, robust, and scalable solution for compressing LLMs without significant performance

degradation and resource consumption.

**Inference cost of the pruned models.** Table 4 presents the statistics of the pruned LLaMA-7B model, including sparsity ratio (SR), number of parameters (Params), MACs, memory consumption, and inference latency. As the sparsity ratio increases, both the model size (Params) and computational requirements (MACs) decrease. For instance, at 33% sparsity, the number of parameters decreases from 6.74B (at 0% sparsity) to 4.52B, and MACs drop from 424.02G to 339.53G, indicating a significant reduction in computational complexity. Similarly, memory usage decreases from 12884.5 MiB to 8718.1 MiB, demonstrating improved memory efficiency. Inference latency is also notably reduced, from 46.95s at 0% sparsity to 34.28s at 33% sparsity (measured on the WikiText2 test set using a single NVIDIA

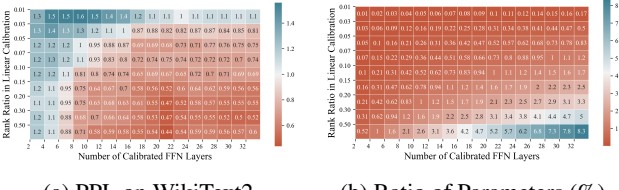

(a) PPL on WikiText2          (b) Ratio of Parameters (%)

*Figure 4.* (a): PPL on WikiText2 datatset (after minus 19 for better presentation), where we vary the number of calibrated FFN layers and the retained ratio of ranks in the linear calibration strategy. (b) each cell represents the ratio (%) of additional parameters introduced by the linear calibration strategy, corresponding to (a).

GeForce RTX 4090). These results show that higher sparsity ratios effectively reduce resource consumption and enhance inference efficiency, making the pruned models more suitable for resource-constrained environments without significant performance loss.

### 4.3. Ablation Study

**Effectiveness of the proposed modules.** In this experiment, we aim to investigate the effectiveness of the proposed Fast-OND and the linear calibration strategy. The experiment is arranged as follows: first, we replace Fast-OND with **standard SVD** (directly performing SVD on both $\mathbf{W}_v$ and $\mathbf{W}_o$), **Activation-Weighted SVD** (performing weighted SVD on both $\mathbf{W}_v$ and $\mathbf{W}_o$ by the magnitudes of input features), and **Wanda** (directly remove neurons based on (5) for $\mathbf{W}_v$ and $\mathbf{W}_o$ ); second, all these experiments are conducted under the scenarios with and without linear calibration. The results are reported in Table 5. We can observe that the proposed Fast-OND is more effective when compared with baselines. Moreover, the linear calibration strategy can be easily coupled with existing pruning methods, and improves their performance.

**Parameters introduced by the linear calibration.** In this experiment, we examine the effect of the number of calibrated FFN layers and the retained ratio $r/d$ for low-rank approximation in linear calibration strategy. From Figure 4 (a), where each cell represents the model performance (measured by PPL) under different scenarios, we observe that both factors affect the PPL. Specifically, when the number of calibrated FFN layers is fixed and the retained ratio increases, the PPL gradually decreases. Similarly, when the retained ratio is fixed and the number of calibrated FFN layers increases, a similar trend is observed. However, the performance gain gradually diminishes once the rank ratio exceeds 0.15 or the number of calibrated FFN layers exceeds 20. In Figure 4 (b), we observe that within the region showing performance gains in Figure 4 (a), the linear calibration strategy introduces very few additional parameters. Specifically, when the number of calibrated FFN layers is 20

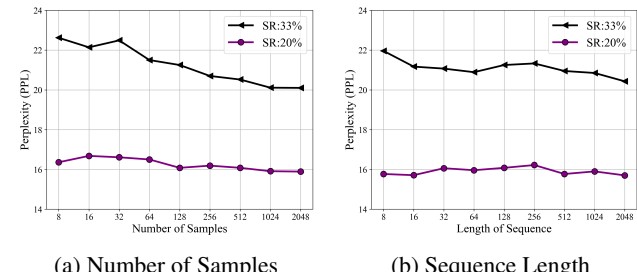

(a) Number of Samples          (b) Sequence Length

*Figure 5.* (a): Vary the number of samples while fixing the length of sequence as 128. (b) Vary the length of sequence while fixing the number of samples as 128. To meet the length of sequence, these samples are sampled from C4 dataset (Raffel et al., 2020).

*Table 6.* Pruning runtime of OND and fast-OND (LLaMA-7B).

| Method | SR | Runtime | PPL ($\downarrow$) | Mean Accuracy ($\uparrow$) |
|---|---|---|---|---|
| OND | 20% | 2910s | **15.17** | 64.32 |
| Fast-OND | 20% | **413s** | 15.35 | **64.54** |

and the retained rank ratio is 0.15, the additional parameters account for around 1%, while the corresponding PPL, as shown in Figure 4 (a), decreases significantly.

### 4.4. Efficiency Analysis

**Sample efficiency.** From Figure 5, we observe that our approach is *not* sensitive to the amount of calibration data. Specifically, both the number of samples and the sequence length vary from 8 to 2048 ($2^8$ times larger), yet the PPL changes by at most 2.4.

**Runtime analysis.** We present the runtime of OND and Fast-OND in Table 6, where we observe that while both OND and Fast-OND achieve similar performance in terms of PPL and accuracy, the runtime of Fast-OND is significantly shorter than that of OND.

### 5. Conclusion

In this paper, we proposed an efficient pruning method for LLMs, called Orthogonal Neuron Decomposition and Linear Calibration (Olica), which eliminates the need for retraining. One of the core designs in the Olica is Orthogonal Neuron Decomposition (OND), which treats the matrix products in the MHA layer as unified entities. Thus, we can use PCA to extract the most information from the MHA layer without sacrificing accuracy or disrupting their original structure. We also devised a fast-OND method, reducing the complexity by a factor of the square of the number of attention heads. In addition, we introduced a linear calibration approach to mitigate the problem of error accumulation in the FFN layer. Using the closed-form solution of ridge

regression, we model the residual errors of the pruned FFN layers with two low-rank matrices, without retraining. Finally, we conducted extensive experiments, showing that the proposed Olica is efficient in terms of data usage, GPU memory, and running time, while delivering superior performance across multiple benchmarks.

## Acknowledgements

The research was partially supported by National Key R&D Program of China (No.2022YFA1003702), National Natural Science Foundation of China (Nos.12426309), New Cornerstone Science Foundation.

## Impact Statement

Our work focuses on structured pruning of LLMs, which improves efficiency by reducing computational and memory requirements, making LLMs more suitable for deployment on resource-constrained devices. The proposed method is efficient in data utilization, GPU memory, and runtime, offering equitable opportunities for the use and development of LLMs. Furthermore, the core idea of Olica—replacing complex structures with simple, explicit expressions—could improve both robustness and interpretability of LLMs, warranting further investigation.

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

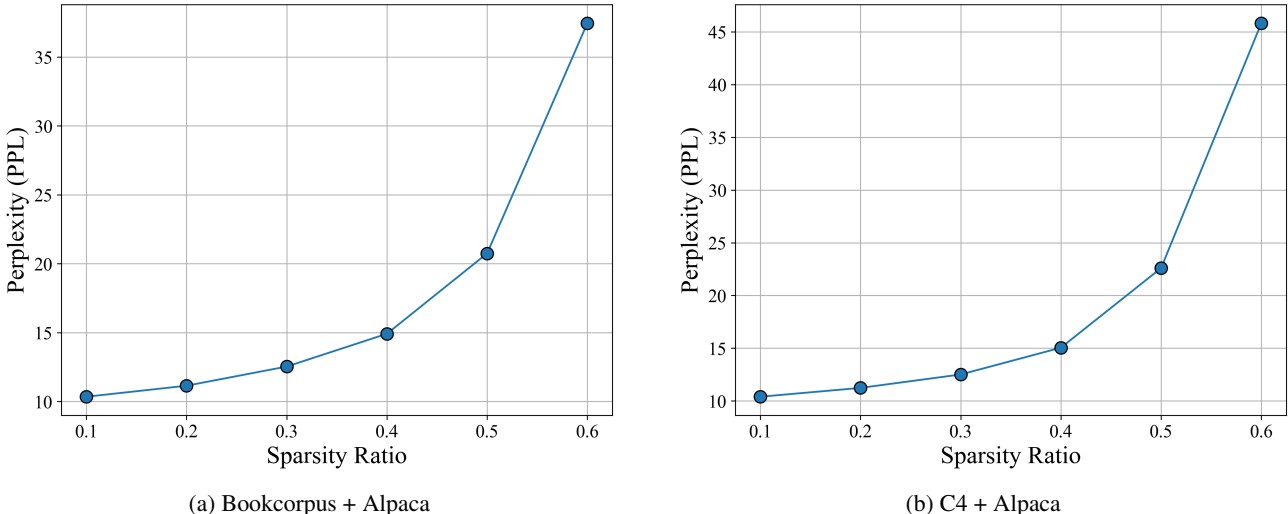

(a) Bookcorpus + Alpaca

(b) C4 + Alpaca

*Figure 6.* Perplexity (PPL) on WikiText2 of LLaMA-30B under different sparsity ratio, where different calibration datasets are used. (a): 256 samples with 128 sequence length are randomly selected from Bookcorpus and Alpaca datasets. (ab): 256 samples with 128 sequence length are randomly selected from C4 and Alpaca datasets.

## A. Implementation Details

We randomly select 128 samples from each of Bookcorpus (Zhu et al., 2015) and Alpaca (Taori et al., 2023) datasets, each of which is truncated to a sequence length of 128 tokens, as the calibration data. The number of calibrated FFN layers is selected from $\{6, 12, 16\}$ for models with different parameter sizes. In the linear calibration, we retain the top 3% eigenvectors for low-rank approximation, i.e., $r/d = 0.03$. Following LoRAP (Li et al., 2024), we compress the MHA and FFN layers, until achieving a specified sparsity ratio of the entire model (i.e., including the token embedding layer and the final projection layer). Following (Frantar & Alistarh, 2023; Frantar et al., 2023), we set the $\lambda$ in 7 as: $\lambda = \lambda_0 \cdot \mathrm{Mean}(\mathrm{diag}(\mathbf{X}^\top \mathbf{X}))$ where $\lambda_0$ is fixed as 0.5.

The form of MHA layer formulated in (2) is one of the most popular transformer architectures. However, the popular open-sourced LLaMA models employ rotary position embedding (RoPE) (Su et al., 2024) in the MHA layer: $\boldsymbol{x}_i \mathbf{W}_q \Phi_{ij} \mathbf{W}_k^\top \boldsymbol{x}_j^\top$, i.e., it inserts a rotary matrix $\Phi_{ij}$ that is based on the position of input variables before the product of $\mathbf{W}_q$ and $\mathbf{W}_k$ to inject the positional information of tokens, which means that there is no direct product between $\mathbf{W}_q$ and $\mathbf{W}_k$. Fortunately, from Figure 2 (c) and (d), we observe significant low-rank property for $\mathbf{W}_q$ and $\mathbf{W}_k$, which means that they are suitable for low-rank approximation. Thus, we apply PCA to $\mathbf{W}_q$ and $\mathbf{W}_k$ respectively, approximating each by two low-rank matrices. The approximation are obtained by retaining the first $r$ principle components. Moreover, inspired by the success of weighted SVD (Hsu et al., 2022; Li et al., 2024), we replace the standard SVD with weighted SVD by the magnitudes of input features with the following objective function:

$$\underset{\mathbf{W}_1, \mathbf{W}_2}{\arg\min} \|(\mathbf{W} - \mathbf{W}_1 \mathbf{W}_2)\mathbf{D}\|_2^2, \tag{12}$$

where $\mathbf{D} = \mathrm{diag}(\|\boldsymbol{x}^{(1)}\|_2, \cdots, \|\boldsymbol{x}^{(d)}\|_2)$ and $\boldsymbol{x}^{(i)}$ is the $i^{th}$ column of input matrix $\mathbf{X}$. This can be solved by performing standard SVD on $\mathbf{W}\mathbf{D} = \mathbf{U}\boldsymbol{\Sigma}\mathbf{V}^\top$: $\mathbf{W}_1 = \mathbf{U}\boldsymbol{\Sigma}$ and $\mathbf{W}_2 = \mathbf{V}\mathbf{D}^{-1}$. Despite performing separate PCA on $\mathbf{W}_q$ and $\mathbf{W}_q$ (this is not our contribution), in the ablation study (i.e., Table 5), we have demonstrated that the Fast-OND is still very important for pruning $\mathbf{W}_v$ and $\mathbf{W}_o$, which highlights the contribution of our proposed Fast-OND.

**Parameter Allocation.** Letting $s$ denotes the sparsity ratio (SR), we first modify the SR as $\hat{s} = s \cdot \frac{M_1}{M_2}$, where $M_1$ is the number of parameters of the entire model and $M_2$ is the number of parameters of all the MHA and FFN layers. Then, the SR for the query and key matrices is $2\hat{s}$; the SR for value and output matrices is $\hat{s}/2$; the SR for the FFN layers is determined by the remaining parameter budgets that ensures the SR of the entire model is $\hat{s}$.

---

**Algorithm 1** Overview of the proposed Olica

---

**Input:** Calibration data $\mathcal{D}$, a model $\mathcal{M}$ (e.g., LLaMA-7B), and the number of FFN layers needed to calibrate $K$, sparsity ratio (SR) $s$;

**Output:** A pruned model $\hat{\mathcal{M}}$

1:  # Step 1: Layer selection
2:  Indexes $\leftarrow$ FFN indexes with $\text{Top}K$ MC2 using (9);
3:  # Step 2: Start pruning
4:  inputs $\leftarrow$ Preprocess($\mathcal{D}$);
5:  **for** block in $\mathcal{M}$.blocks **do**
6:      outputs $\leftarrow$ block(inputs);
7:      # Pruning MHA layer with SR $s$
8:      $\mathbf{W}_q, \mathbf{W}_k, \mathbf{W}_v, \mathbf{W}_o \leftarrow$ block.MHA.matrices;
9:      $\hat{\mathbf{W}}_q, \hat{\mathbf{W}}_k \leftarrow \text{WeightedSVD}(\mathbf{W}_q, \mathbf{W}_k)$;
10:     $\hat{\mathbf{W}}_v, \hat{\mathbf{W}}_o \leftarrow \text{Fast\_OND}(\mathbf{W}_v, \mathbf{W}_o)$;
11:     # Pruning and calibrating FFN layer with SR $s$
12:     $\hat{f}_{\text{FFN}} \leftarrow$ using (5);
13:     **if** block.FFN.index in Indexes **then**
14:         $\hat{\mathbf{W}}_1, \hat{\mathbf{W}}_2 \leftarrow$ Performing SVD on the solution of (7);
15:         Modifying block.FFN.forward as (11);
16:     **end if**
17:     inputs $\leftarrow$ outputs;
18: **end for**
19: **return** The pruned model $\hat{\mathcal{M}}$

---

## B. Algorithm.

**Algorithm.** As an example, we use LLaMA models to present the final algorithm in Algorithm 1.

## C. Pruning Results of LLaMA-30B

Here, we provide the pruning results of a larger model, i.e., LLaMA-30B. The results for the pruned LLaMA-30B model are presented in Figure 6. We report results for sparsity ratios of 0.1, 0.2, 0.3, 0.4, 0.5, and 0.6. Additionally, we evaluate different combinations of datasets for calibration data, namely Bookcorpus + Alpaca and C4 + Alpaca. It can be observed that without retraining, the LLaMA-30B model can tolerate a maximum sparsity ratio of 40%. Beyond this threshold, when the model is pruned with a higher sparsity ratio, its performance deteriorates significantly. Furthermore, when the sparsity ratio is below 40%, the choice of calibration dataset has a minimal impact on the pruning results.

## D. Results of 50% Sparsity Ratio

We observe that at high sparsity ratios (e.g., 50%), the performance of pruned LLMs deteriorates significantly. For instance, in the results reported by (Ma et al., 2023; Li et al., 2024; Ling et al., 2024), the performance of the pruned LLaMA-13B model with a 50% sparsity ratio is notably lower than that of LLaMA-7B, even with retraining. This suggests that a 50% sparsity ratio may exceed the tolerance limit of these models. However, we believe that when pursuing a high sparsity ratio, the primary prerequisite is to maximally maintain the performance of LLMs. Therefore, we provide the results of LLMs with 50% sparsity ratio in the Appendix.

Table 7 presents the zero-shot performance and perplexity (PPL) results on the WikiText2 dataset for various models with a 50% sparsity ratio. The table compares the performance of different pruning methods, including Olica (our proposed method), LLM-Pruner, and LoRAP.

- LLaMA-7B: At 50% sparsity, Olica achieves an average accuracy of 50.68%, which is higher than both LLM-Pruner (48.35%) and LoRAP (46.86%). Olica also performs well on BoolQ (65.32%) and ARC-e (51.98%), with a competitive PPL of 49.39. This demonstrates that Olica balances compression with high performance across several reasoning tasks.

*Table 7.* Zero-shot performance and PPL on wikitext2 with 50% sparsity ratio.

| Model | Method | FT | PPL (↓) | BoolQ | PIQA | HellaS | WinoG | ARC-e | ARC-c | OBQA | Avg. (↑) |
|---|---|---|---|---|---|---|---|---|---|---|---|
| LLaMA-7B | Dense | ✗ | 12.19 | 77.71 | 79.05 | 76.00 | 68.98 | 74.58 | 46.33 | 44.20 | 66.69 |
| | LLM-Pruner | ✓ | 40.64 | 60.21 | **68.88** | 47.86 | 54.62 | 43.94 | 27.73 | **35.20** | 48.35 |
| | LoRAP † | ✗ | 54.19 | 53.85 | 64.74 | **48.58** | 57.30 | 44.28 | 27.30 | 32.00 | 46.86 |
| | Olica (Ours) | ✗ | **49.39** | 65.32 | 66.87 | 47.80 | **58.72** | **51.98** | **30.72** | 33.40 | **50.68** |
| LLaMA-13B | Dense | ✗ | 11.58 | 77.89 | 80.14 | 79.06 | 72.85 | 74.75 | 47.61 | 44.80 | 68.16 |
| | LLM-Pruner | ✓ | 74.62 | 62.35 | **72.74** | 58.43 | 55.88 | 51.89 | 33.02 | **38.20** | 53.22 |
| | LoRAP † | ✗ | **34.93** | 57.68 | 70.24 | **59.93** | 62.19 | 54.46 | 31.83 | 36.00 | 53.18 |
| | Olica (Ours) | ✗ | 35.35 | **70.03** | 71.16 | 58.63 | **65.35** | **61.41** | **34.47** | 33.80 | **56.41** |
| LLaMA2-7B | Dense | ✗ | 12.19 | 77.71 | 79.05 | 76.00 | 68.98 | 74.58 | 46.33 | 44.20 | 66.69 |
| | LoRAP † | ✗ | 70.33 | 51.87 | 62.08 | **43.78** | 57.54 | 40.78 | 26.79 | **32.20** | 45.00 |
| | Olica (Ours) | ✗ | **52.82** | **64.07** | **62.73** | 43.34 | 53.91 | **49.49** | **28.24** | 31.20 | **47.57** |
| LLaMA2-13B | Dense | ✗ | 10.98 | 80.55 | 80.52 | 79.37 | 72.21 | 79.38 | 48.98 | 45.20 | 69.46 |
| | LoRAP † | ✗ | 37.55 | **67.13** | 69.04 | **55.60** | 58.96 | 53.83 | 31.14 | **34.00** | 52.81 |
| | Olica (Ours) | ✗ | **34.21** | 65.96 | **70.67** | 54.63 | **60.30** | **60.31** | **34.04** | 33.60 | **54.21** |
| Vicuna-7B | Dense | ✗ | 16.11 | 78.41 | 78.56 | 74.68 | 70.09 | 72.01 | 43.77 | 43.40 | 65.85 |
| | LLM-Prunerr | ✓ | **43.96** | 40.76 | **67.08** | 46.64 | 53.28 | 43.98 | 27.56 | **34.00** | 44.76 |
| | LoRAP † | ✗ | 82.18 | 48.62 | 63.71 | 47.26 | 55.96 | 42.34 | 28.33 | 31.00 | 45.31 |
| | Olica (Ours) | ✗ | 56.35 | **56.02** | 66.27 | **51.11** | **56.99** | **52.31** | **31.91** | 32.20 | **49.54** |
| Vicuna-13B | Dense | ✗ | 13.50 | 85.29 | 79.11 | 77.51 | 71.59 | 78.66 | 50.77 | 45.40 | 69.76 |
| | LoRAP † | ✗ | 45.58 | 70.61 | 69.10 | **58.15** | 60.62 | 56.27 | 34.04 | **36.40** | 55.02 |
| | Olica (Ours) | ✗ | **39.22** | **72.02** | **70.29** | 57.59 | **61.80** | **62.16** | **36.95** | 35.20 | **56.57** |

- LLaMA-13B: For this larger model, Olica again outperforms LLM-Pruner and LoRAP, with an average score of 56.41%, which is higher than both methods (LLM-Pruner: 53.22%, LoRAP: 53.18%). Olica also delivers superior performance on datasets like BoolQ (70.03%) and WinoG (65.35%), with a relatively low PPL of 35.35.

- LLaMA2-7B: Olica shows strong performance with an average score of 47.57%, significantly surpassing LoRAP (45.00%) and providing solid results across the tasks. It achieves a low PPL of 52.84.

- Vicuna-7B: Olica performs well here too, with an average score of 49.54%, surpassing LLM-Pruner (44.76%) and LoRAP (45.31%). It also maintains a lowest PPL of 56.35.

- Vicuna-13B: At 50% sparsity, Olica outperforms LoRAP (55.02%) with an average score of 56.57%. It achieves a low PPL of 39.22, showing the robustness of Olica across larger models as well.

In summary, Olica consistently outperforms both LLM-Pruner and LoRAP across all tested models, achieving better or comparable performance while maintaining competitive perplexity. These results demonstrate the effectiveness of Olica as a pruning method that preserves model performance while significantly reducing model size and computational complexity.

