# OpenReview forum: "Olica: Efficient Structured Pruning of Large Language Models without Retraining"
_ICML.cc/2025/Conference — ICML 2025 poster_

### Official Review · Reviewer_uBxH · 2025-03-07

**Overall Recommendation:** 4

**Summary:**

This work proposes using PCA to compress the matrix product in MHA with a fast computation approach for LLM compression. Additionally, to address errors caused by pruning, a reconstruction method based on ridge regression is introduced for FFN. The experiments cover LLaMA-based models.

**Claims And Evidence:**

I think the claims of this paper are well supported by clear and compelling evidence. This work introduces a computationally efficient pruning method for compressing LLMs. The concept of applying PCA to matrix products in MHA is interesting, and further performing SVD on one of the product matrices for fast computation appears effective. I appreciate the interesting comparison in Tables 5 and 6.

**Essential References Not Discussed:**

* In terms of FFN pruning, several retraining-free methods (e.g., A, B, and C below) have already been proposed. Particularly, the concept of reconstruction recovery, defined as reducing the difference between activations from the original weights and those from compressed weights, is widely applied these days (e.g., B and C). Could you explain why the proposed calibration method is considered superior to existing methods?
  - [A] A Fast Post-Training Pruning Framework for Transformers https://arxiv.org/abs/2204.09656
  - [B] Gradient-Free Structured Pruning with Unlabeled Data https://arxiv.org/abs/2303.04185
  - [C] Fluctuation-based Adaptive Structured Pruning for Large Language Models https://arxiv.org/abs/2312.11983

* This work concentrates on the width pruning of LLMs. However, it appears to lack a thorough discussion or comparison with other pruning methods: width pruning (FLAP, Minitron), depth pruning (ShortGPT, Shortened LLaMA, SLEB, Minitron), and hybrid width-depth pruning (Sheared LLaMA). A comparison with FLAP, in particular, seems essential given that the primary advantage of this work is its retraining-free approach, and the concept of reconstruction loss recovery looks similar.
  - [FLAP] https://arxiv.org/abs/2312.11983
  - [ShortGPT] https://arxiv.org/abs/2403.03853
  - [Shortened LLaMA] https://arxiv.org/abs/2402.02834
  - [SLEB] https://arxiv.org/abs/2402.09025
  - [Sheared LLaMA] https://arxiv.org/abs/2310.06694
  - [Minitron] https://arxiv.org/abs/2408.11796

**Experimental Designs Or Analyses:**

* I am a bit unclear why OND and FastOND outperform SVD and AWSVD in Table 5. In my understanding, SVD and AWSVD treat each of W_v and W_o independently, which might provide more accurate estimates than treating the product of the two matrices, W_vW_o^T. Could you provide additional explanation and/or analysis for this?
* Why is the importance score for the gate projection of the FFN missing in Equation (5)? How is the pruning of the FFN’s gate projection handled in the LLaMA family (e.g., is it pruned in the same manner as the up projection)? Could you also elaborate on the differences and merits of this approach compared to LoRAP?
* The pruning ratios explored in this work appear relatively low. Is this approach still effective at higher pruning ratios, for instance, exceeding 50%? Based on my experience, pruning only 20-30% of weights typically does not yield significant computational efficiency gains. Therefore, I personally prefer the approach of extreme compression (beyond 50%), along with subsequent retraining, as it is more effective for achieving notable speedups and substantial memory reductions. Could you provide additional experimental results or insights on this topic?
* The models used for experimental validation in this work appear quite limited, as they are confined to the LLaMA family. I would recommend including experimental comparisons with other models (e.g., OPT, Qwen, and Phi) and MoE-based architectures alongside existing methods. This would help verify the broad applicability and superiority of this work.
* I think the methodology for calculating latency gains in Table 4 should be described in more detail, as width pruning has been reported to be challenging for achieving actual speedups in certain setups (e.g., ZipLM, Shortened LLaMA). I suspect that the speedup reported in Table 4 may result from measurements taken only during the prefill stage, excluding decoding, and possibly under a large batch size. Could you specify what kind of framework (such as Vanilla HuggingFace, VLLM, etc.), batch size (1 or more), and output generation length were used? Additionally, can this method achieve speedups with a batch size of 1 for both prefill and decoding stages?
  - [ZipLM] https://arxiv.org/abs/2302.04089
  - [Shortened LLaMA] https://arxiv.org/abs/2402.02834
* I believe that the baseline comparison could be enhanced by including the studies I outlined in the section 'Essential References Not Discussed' below.

**Methods And Evaluation Criteria:**

I believe the proposed method and setup are well-aligned with previous LLM pruning studies. Although numerous methods exist for evaluating LLMs, considering this work's focus on compression, employing metrics such as PPL along with LLaMA/LLM-Pruner's benchmarks appears to be sufficient.

**Other Comments Or Suggestions:**

N/A

**Other Strengths And Weaknesses:**

* Proper analyses have been conducted with in-depth ablation studies, though the experiments cover solely the LLaMA family.
* Overall, the paper is well-written and easy to follow.

**Questions For Authors:**

Please refer to <Experimental Designs Or Analyses> and <Essential References Not Discussed>.

I appreciate the authors' efforts to compress LLMs with reduced computation and generally like the main idea and flow of this paper. However, my overall opinion of this work is borderline, primarily due to some unclear aspects of the results and method, coupled with limited experimental validation. While I do not favor reviews that demand excessive experiments (often appearing to seek reasons for rejection), I believe that additional results could significantly strengthen the value of this work. Given that light computation for compression is a major advantage of this work, extending the experiments would be feasible. I hope my comments will contribute to enhancing the impact of this study.

**Relation To Broader Scientific Literature:**

I think compressing LLMs, particularly with reduced computation, is a hot topic, and this work has done a good job addressing it. Several ablations presented in this work demonstrate that naively applying SVD-based methods does not perform well, while employing a Wanda-based importance criterion and considering matrix products appears effective. Although the concept of FFN compression with reconstruction-based loss seems quite similar to previous work, the further developments with layer selection and low-rank approximation are interesting.

**Theoretical Claims:**

I believe this paper clearly presents the method in general, including matrix products for MHA and related SVD-based methods, as well as regression-based FFN compression.

---

> ### Author Rebuttal · Authors · 2025-04-01
>
> We really  appreciate the time and efforts you extended in reviewing our paper. Below please find our responses regarding your concerns.
>
> **Q1**: SVD and AWSVD treat each of $W_v$ and $W_o$ independently, which might provide more accurate estimates than treating the product of the two matrices, $W_vW_o^T$.
>
> **A1**: Thanks for your question. The reconstruct loss $|| XW_{vo} - X\hat{W_{vo} }||$ relies primarily on $W_{vo}=W_vW_o^{\top}$, rather than each $W_v$ and $W_o$ individually. Although $W_v$ and $W_o$ can be individually estimated accurately by SVD and AWSVD, the reconstruction loss of  the product $W_vW_o^T$ can not be guaranteed, as it may incur the error accumulation issue introduced by the product of separately estimated $\hat{W_v}$ and $\hat{W_o}$.
>
> **Q2:** Is the gate projection pruned in the same manner as the up projection? What are the differences compared to LoRAP?
>
> **A2:** Yes, the gate projection of the FFN layer is pruned in the same manner as the up projection and this is coincide with LoRAP. Our purpose here is to design the linear calibration, i.e., approximating the residual errors $E=f(X)-\hat{f}(X) \approx X\hat{W}$ by a linear model, such that $f(X)\approx \hat{f}(X)+ X\hat{W}$, no matter how $f$ is pruned to $\hat{f}$. In principle, any efficient pruning methods for $f$ can be adopted here.
>
> **Q3:** Could you provide additional experimental results on higher pruning ratio?
>
> **Table 1:** Results of larger models with 50% sparsity ratios.
> |Method|PPL|Avg. Acc.|
> |-------|-------|-------|
> |LLaMA-30B (dense)|9.8| 72.1|
> |LoRAP|23.4| 60.9 |
> |Olica (Ours) |**20.9**|**63.6**|
> |LLaMA-2-70B (dense)|9.2 |73.6|
> |LoRAP|16.9|64.7|
> |Olica (Ours)|**14.8**|**67.3**|
>
> **A3:** We have demonstrated the results of 50\% sparsity ratio in the Appendix D of our paper, including 7B and 13B models. Here, we further scale the model parameters up to 30B and 70B. From Table 1, we can observe that our proposed method still works better than LoRAP under higher pruning ratio and larger models. As for the fine-tuned results of these pruned larger models, we leave to future works due to the resource constraints.
>
> **Q4:** Results of other models (e.g., OPT, Qwen, and Phi) and MoE-based architectures.
>
> **Table 2:** Results of 20% sparsity ratio.
>  |Method|PPL (A)| Avg. Acc. (A)|PPL (B)| Avg. Acc. (B)| PPL (C)| Avg. Acc. (C)|
> |-------|-------|-------|-------|-------|-------|-------|
> |Dense|23.73|69.58 |11.91|73.18|9.33|74.49|
> |LoRAP|42.78| 58.57|25.34|64.23|12.87|69.21|
> |Olica (Ours)| **39.77**|**61.78**|**21.15**|**66.67**|**11.28**|**70.95**|
>
> **A4:** We further extended the evaluation results to LLMs Phi-2 (A), Qwen2.5-14B (B), and Mixtral-8x7B-v0.1 (C), where Mixtral-8x7B-v0.1 is a MoE-based model. As shown in Table 2, we see that our approach consistently outperforms LoRAP, showing high applicability and superiority.
>
> **Q5:** What are the details of calculating the latency? Can this method achieve speedups with a batch size of 1 for both prefilling and decoding stages?
>
> **A5:** Indeed, following the evaluation protocols of works LLM-Pruner and LoRAP, we mainly tested the speedup of the prefilling state, where the vanilla HuggingFace is used and the batch size is set to 2. As for the decoding stage, please see Table 3 in the **Q4** raised by **Reviewer 9AzP**, we can see that under 50% sparsity, the throughput in tokens of pruned model can enhance about 30\%.
>
> **Q6:**  Could you explain why the proposed calibration method is considered superior to A, B, and C?
>
> **A6:** Both A and B propose to adjust the combination of the remaining filters to minimize the reconstruction error. However, if the pruned filters contain information not carried by any remaining filters, this adjustment is insufficient. Our proposed approach directly models the residual errors, complementing the remaining filters by introducing new information they do not capture. As for C, it uses a constant baseline to compensate for the pruned filters, which may reduce the accuracy. As shown in Table 3  presented in **A7**,  FLAP performs significantly worse than the proposed Olica.
>
>
> **Q7:** Comparisons with Shortened LLaMA, FLAP, and SLEB (LLaMA-7B).
>
> **Table 3:**  Comparison results.
> |Method |PPL |Avg. Acc.|
> |-------|-------|-------|
> |FLAP|17.0|59.5|
> |SLEB|18.5|57.6|
> |Shortaned LLaMA: Grad+|20.2|63.5|
> |Shortaned LLaMA: PPL|17.7|61.9|
> |Olica (Ours)|**15.4**|**64.5**|
>
> **A7:**  From Table 3, we see that our proposed Olica consistently outperforms these baselines. We do not compare with methods Sheared LLaMA and Minitron due to the unfairness, as they require huge computation and data resources to retrain the pruned model. For example, to retrain a 2.7B model, Sheared LLaMA requires 50B tokens and 16 GPUs. In contrast, our proposed method is highly efficient, enabling pruning models with more than 70B parameters on a single NVIDIA GeForce RTX 4090 GPU with less than an hour runtime.

---

> > ### Comment · Reviewer_uBxH · 2025-04-08
> >
> > I appreciate the clear answers (especially for A1 and A2) and the additional experiments provided. Although the 50% pruning ratios show worse PPL results, Olica appears competitive compared to LoRAP. Furthermore, I think the scaling up to 30B and 70B models (A3) needs to be acknowledged. Additional experiments compared to depth pruning methods (A7) are also well-presented.
> >
> > I also agree that a direct comparison to Sheared LLaMA and Minitron would be unfair or impossible due to limited computing and time. Furthermore, I feel the additional experiments over different models confirm the general applicability of this work (A4). Thank you for detailing the speedup experiments (A5).
> >
> > This work was initially positive among the LLM pruning papers I was assigned, and after the rebuttal, my view has become even more favorable. The concept of applying PCA to matrix products in MHA is interesting and I believe it can facilitate future research. Based on the sincere rebuttal, the innovative approach, and the extensive experiments, I would like to update my score from 3 to 4.

---

> > > ### Author Response · Authors · 2025-04-09
> > >
> > > We're glad to hear that the additional experiments and our proposed approach were favorably evaluated. We really appreciate your valuable feedback, supportive perspective, and the revised evaluation.

---

### Official Review · Reviewer_9AzP · 2025-03-13

**Overall Recommendation:** 2

**Summary:**

This paper proposes Olica, a structured pruning method for large language models that eliminates the need for retraining. The approach introduces Orthogonal Neuron Decomposition to compress the multi-head attention layer using PCA-based factorization and Linear Calibration to mitigate pruning-induced errors in feed-forward networks using ridge regression. Experimental results suggest that Olica achieves competitive performance while reducing computational costs.

**Claims And Evidence:**

The paper emphasizes that Olica eliminates retraining, but the linear calibration step implicitly fine-tunes the model using calibration data. While this is not full retraining, it is still an optimization step requiring data, making the claim of "no retraining" misleading.
The authors compare Olica against pruning methods that explicitly use LoRA fine-tuning, but they do not evaluate against one-shot pruning methods that also do not retrain the model (e.g., magnitude-based or Taylor expansion-based structured pruning).

Some claims, such as "significantly reduces running time while delivering better performance," are not statistically verified. Confidence intervals or significance tests should be provided.

**Essential References Not Discussed:**

Some recent works are missing. For example, "Search for Efficient Large Language Models" published in NeurIPS 2024. Please compare with works published in NeurIPS 2024 or ICLR 2025 if possible.

**Experimental Designs Or Analyses:**

I check the soundness/validity of all experimental designs or analyses.

**Methods And Evaluation Criteria:**

It makes sense for the application.

**Other Comments Or Suggestions:**

Please refer to my previous comments.

**Other Strengths And Weaknesses:**

**Strengths:**

1. The authors conduct extensive experiments across multiple LLMs and benchmarks, comparing Olica against state-of-the-art methods.

**Weaknesses:**

1. The paper emphasizes that Olica eliminates retraining, but the linear calibration step implicitly fine-tunes the model using calibration data. While this is not full retraining, it is still an optimization step requiring data, making the claim of "no retraining" misleading.
The authors compare Olica against pruning methods that explicitly use LoRA fine-tuning, but they do not evaluate against one-shot pruning methods that also do not retrain the model (e.g., magnitude-based or Taylor expansion-based structured pruning).

2. All experiments are conducted on LLaMA-7B and 13B. The scalability of Olica to models with 30B+ parameters is not demonstrated.
The pruning ratios tested (up to 33% sparsity) are moderate. In real-world applications, structured pruning often targets higher sparsity ratios (e.g., 50–75%), and the performance at these levels is unknown.

3. The results rely on WikiText-2 and a small set of multiple-choice benchmarks. There is no evaluation on open-ended tasks, instruction following, or reasoning-heavy datasets, where pruning might degrade model coherence.
While FLOP reductions are reported, actual inference speedup (e.g., throughput in tokens per second) is not benchmarked on real-world workloads.
No ablations compare Olica against simpler structured or semi-structured pruning baselines.

5. Several key ideas (e.g., how ridge regression is applied) are underexplained. The methodology section assumes familiarity with pruning literature but does not adequately define important concepts for a broad ICML audience.
Some claims, such as "significantly reduces running time while delivering better performance," are not statistically verified. Confidence intervals or significance tests should be provided.

**Questions For Authors:**

Please refer to my previous questions.

**Relation To Broader Scientific Literature:**

There is no contribution of the paper related to the broader scientific literature. This paper focus on the application aspects.

**Theoretical Claims:**

There is no theoretical claim in this paper.

---

> ### Author Rebuttal · Authors · 2025-04-01
>
> We greatly thank you for the detailed reviews and helpful suggestions. We reply point-by-point here.
>
> **Q1:** The claim of "no retraining" is misleading.
>
>  **A1:** Thanks for your question. In deep learning,  "training'' or "fine-tuning'' generally require a series of forward and backward processes to compute the gradients so as to update the parameters of neural networks. However, Our Olica only takes one forward process and does not involve the  backward process. In this sense, our method does not require retraining. In the literature [1, 2], they also refer this paradigm as **no retraining**.
>
> [1] Sparsegpt: Massive language models can be accurately pruned in one-shot. ICML, 2023.
>
> [2] A simple and effective pruning approach for large language models. ICLR, 2024.
>
>
> **Q2**: Comparions to one-shot magnitude-based or Taylor expansion-based structured pruning.
>
> **A2**: Sorry for the confusion. We  clarify that the baselines **LLM-Pruner** and **LoRAP** are one-shot Taylor expansion-based and magnitude-based methods, respectively. Their results have been included in Table 2 and Table 3 of  our paper.
>
> **Q3:** The performance of 30B+ model and higher pruning ratio are unknown.
>
> **Table 1**: Results of larger models and higher sparsity ratios (SR), where A=LLaMA-30B, B=LLaMA-2-70B.
> | Model (SR) | PPL ||  Avg. Acc. ||
> |-------|-------|-------|-------|-------|
> | A (0%) | 9.8 | | 72.1| |
> |  | LoRAP | Olica |LoRAP | Olica |
> | A (20%)  | 11.6 | **11.1** | 70.3 |**71.1**|
> | A (50%)  | 23.4 | **20.9** | 61.9 |**63.6**|
> | B (0%) | 9.2 | | 73.6| |
> | B (20%)  | 9.9 | **9.2**| 71.3 | **72.4**|
> | B (50%)  | 16.3 |**14.8**|65.1|**67.3**|
>
> **A3:** We further conducted experiments on models LLaMA-30B and LLaMA-2-70B. As shown in Table 1, we see that our proposed Olica consistently achieves better performance with different pruning raios and model scales,  which clearly demonstrates the scalability of our porposed Olica.
>
> **Q4**: No evaluation on challenging datasets. Throughput in tokens per second is not benchmarked.
>
> **Table 2:** Results (accuracy) on MMLU of Llama-2-13b-chat (20% sparsity).
> |Method|Humanities|Social Sciences |STEM|Other|Avg.|
> |-------|-------|-------|-------|-------|-------|
> |Dense|49.5|62.1|44.0|59.9|53.9|
> |LoRAP|41.2|49.7|36.8|48.9|44.2|
> |Olica|**43.7**|**52.5** |**38.2**|**50.8**|**46.3**|
>
> **Table 3:** Throughput in tokens per second of pruned LLaMA-30B tested on 3 RTX 4090 GPUs using vanilla HuggingFace.
>
>  |Sparsity |0%| 20%| 30%| 50|
> |-------|-------|-------|-------|-------|
> |Tokens/s |14.2|16.2|16.9|18.3|
>
> **A4:** We exend the evaluation results to Massive Multi task Language Understanding (MMLU) task, which is a quiz bank covering 57 subjects, presenting a greater challenge compared to the Commonsense Reasoning datasets. From Table 2, we can still obtain better performance. We further tested the throughput in tokens of pruned LLaMA-30B. As shown in Table 3,  under 50% sparsity, the throughput in tokens of pruned model can enhance about 30%.
>
> **Q5:** Several key ideas, e.g., how ridge regression is applied, are underexplained.
>
> **A5**: As for the ridge regression, since our target is to estimate the the pruning errors by a linear model: $E=f(X)-\hat{f}(X) \approx X\hat{W}$, where $\hat{f}$ is the pruned version of $f$ and $\hat{W}$ is the parameter matrix to estimate, a natural solution
> is the least-square estimation: $\hat{W}=(X^{\top}X)^{-1}E$. However, in the modern LLMs, the input matrix $X$ is extremely high-dimensional, leading to  the irreversibility of $X^{\top}X$. To solve the problem, following the ridge regression, we add a $l_2$ penalty to the regression loss: $ ||E - XW|| + \lambda || W  ||^{2}_{F}$. This leads
> a closed-form solution: $\hat{W}=(X^{\top}X+\lambda I)^{-1}E$, where $I$ is an identity matrix.
>
> **Q6:** Confidence intervals or significance tests should be provided.
>
> **Table 4:** t-test of comparison experiments of our paper.
>  |Experiments| LLaMA-7B (20%) | LLaMA-7B (25%) | LLaMA-7B (33%) | LLaMA-13B (20%) | LLaMA-2-7B (30%) | Vicuna-7B (20%) |
> |-------|-------|-------|-------|-------|-------|-------|
> | p-value | 2.221$\times 10^{-5}$ |6.358$\times 10^{-5}$ | 2.368$\times 10^{-6}$|2.093$\times 10^{-4}$| 2.943$\times 10^{-8}$| 6.036$\times 10^{-4}$|
>
> **A6:**   To conduct significance tests, we set the null hypothesis as "the baseline methods and the proposed Olica have the same accuracy $\mu$".  We conduct five random experiments by independently selecting calibration datasets, over which we record the mean accuracy, denoted as $\hat{\mu}$. The significance tests of p-values are reported in Table 4, and we can see that all the null hypotheses are rejected when p-value $<$ 0.01.
>
> **Q7:** Some recent works are missing. For example, [1] Search for Efficient Large Language Models.
>
> **A7:** We directly cite the results from [1]. Under the 20% sparsity, our average accuracies are: 61.10% (LLaMA-7B) and 63.73% (LLaMA-13B), whereas [1] are 59.71%  (LLaMA-7B) and 62.10% (LLaMA-13B).

---

### Official Review · Reviewer_ZyT3 · 2025-03-14

**Overall Recommendation:** 2

**Summary:**

Olica is a retraining-free structured pruning framework for Large Language Models (LLMs), with orthogonal decomposition and linear calibration. It unifies MHA matrix products and applies PCA to preserve essential information while compressing the model. A fast decomposition method reduces PCA complexity, and a linear calibration technique reconstructs residual errors in pruned FFN layers using SVD.

**Claims And Evidence:**

1.	The key observation of this article is that MHA depends on W_q * W_k^T, which is not true on llama, because of RoPE. However, all experiments in this article are done on llama. This means that all the discussion in Section 3.2 doesn’t apply to llama. In Appendix A, authors claim that they apply PCA separately on W_q and W_k. This part is quite unclear. And If PCA is separately applied on W_q and W_k, what is the innovation of this compared with SVD-LLM [1], ASVD [2]?
2.	The linear calibration turns the skip connection into weighted low-rank layer, which is quite similar to SliceGPT. SliceGPT also converts the skip connection into a multiplication of two low-rank matrices. What is the contribution of this part?
3.	In paragraph of “Fast OND”, authors claim that Figure 2 shows that the low-rank structure of the product W_v W_o^T can be determined by one of the W_v and W_o. But Figure 2 only shows that the trend of singular value in W_v and W_o but doesn’t show that they have similar singular vectors. So the motivation of performing SVD on one of W_v and W_o seems not very solid.

[1] Wang, Xin, et al. "Svd-llm: Truncation-aware singular value decomposition for large language model compression." arXiv preprint arXiv:2403.07378 (2024).
[2] Yuan, Zhihang, et al. "Asvd: Activation-aware singular value decomposition for compressing large language models." arXiv preprint arXiv:2312.05821 (2023).

**Essential References Not Discussed:**

[1] Wang, Xin, et al. "Svd-llm: Truncation-aware singular value decomposition for large language model compression." arXiv preprint arXiv:2403.07378 (2024).
[2] Yuan, Zhihang, et al. "Asvd: Activation-aware singular value decomposition for compressing large language models." arXiv preprint arXiv:2312.05821 (2023).

**Experimental Designs Or Analyses:**

yes

**Methods And Evaluation Criteria:**

yes

**Other Comments Or Suggestions:**

1.	Line 25, “extracte” -> “extract”

**Other Strengths And Weaknesses:**

no

**Questions For Authors:**

1. If PCA is separately applied on W_q and W_k, what is the innovation of this compared with SVD-LLM [1], ASVD [2]?
2.	The linear calibration turns the skip connection into weighted low-rank layer, which is quite similar to SliceGPT. SliceGPT also converts the skip connection into a multiplication of two low-rank matrices. What is the contribution of this part?
3.	In paragraph of “Fast OND”, authors claim that Figure 2 shows that the low-rank structure of the product W_v W_o^T can be determined by one of the W_v and W_o. But Figure 2 only shows that the trend of singular value in W_v and W_o but doesn’t show that they have similar singular vectors. So the motivation of performing SVD on one of W_v and W_o seems not very solid.

**Relation To Broader Scientific Literature:**

This article proposes a new way of retraining-free structured pruning, which could be benefit for model compression.

**Theoretical Claims:**

yes

---

> ### Author Rebuttal · Authors · 2025-04-01
>
> Thank you for providing the insightful comments. We will try our best to address your concerns as follows.
>
> **Q1:** The key observation of this article is that MHA depends on $W_{q_i}W_{k_i}^{\top}$, which is not true on llama, because of RoPE. However, all experiments in this article are done on llama. This means that all the discussion in Section 3.2 doesn’t apply to llama. In Appendix A, authors claim that they apply PCA separately on $W_q$ and $W_k$. This part is quite unclear. And If PCA is separately applied on $W_q$ and $W_k$, what is the innovation of this compared with SVD-LLM [1], ASVD [2]?
>
> **A1:** Despite the separate estimation of $W_q$ and $W_k$, it cannot diminish our contribution to the estimation of $W_v$ and $W_o$.
> Our approach is different from both SVD-LLM and ASVD in the following aspects.  First, both SVD-LLM and ASVD treat the matrices $W_v$ and $W_o$ independently and propose different variants of SVD to reconstruct $W_v$ and $W_o$ individually.
> Considering that  the reconstruction loss  $||XW_{vo} - X\hat{W_{vo}}||$ relies on $W_{vo}=W_vW_o^{\top}$, rather than each $W_v$ and $W_o$ individually, the proposed Olica regards the product $W_{vo}=W_vW_o^{\top}$ as an unified entity. Although
> SVD-LLM and ASVD can gain more accurate estimations of  $W_v$ and $W_o$, the reconstruction loss of the product  $W_{vo}$ can not be guaranteed, because  it may incur the error accumulation issue induced by the product of separately estimated $\hat{W_{v}}$ and $\hat{W_{o}}$. As evidenced by Table 5 of our paper, the proposed method can achieve significantly better performance than separate estimations of $W_v$ and $W_o$. Moreover, we  compare the proposed Olica with SVD-LLM based on the same evaluation settings.  We obtain the following performance on three datasets: ARC\_easy (69\%), PIQA (78\%) and WinoG (70\%) using Olica, whereas the performance of SVD-LLM(W) is: ARC\_easy (62\%),  PIQA (71\%), and  WinoG  (61\%). These results clearly demonstrated the advantages of our proposed method.
>
> [1] Wang, Xin, et al. "Svd-llm: Truncation-aware singular value decomposition for large language model compression." arXiv preprint arXiv:2403.07378 (2024).
>
> [2] Yuan, Zhihang, et al. "Asvd: Activation-aware singular value decomposition for compressing large language models." arXiv preprint arXiv:2312.05821 (2023).
>
> **Q2:** The linear calibration turns the skip connection into weighted low-rank layer, which is quite similar to SliceGPT. SliceGPT also converts the skip connection into a multiplication of two low-rank matrices. What is the contribution of this part?
>
> **A2:**  Thanks for your insightful comments. The proposed linear calibration (LC)  is designed to calibrate the residual information of pruned layers by linear models, that is, $E=f(X)-\hat{f}(X)\approx X\hat{W}$, such that the pruned layer $\hat{f}$ can be calibrated to approximate its original version: $f(X) \approx \hat{f}(X) + X\hat{W}$. To further reduce the number of parameters of $\hat{W}$, we performed SVD on $\hat{W}$. The SliceGPT inserts an identity matrix $I=UU^{\top}$ into the intermediate layers of transformer so that to reduce its hidden dimension, where the orthogonal matrix $U$ is obtained by perform SVD on the feature matrix $X$. We emphasize that the SVD is a general technique and can be used for different purposes. Here, we use it for reducing the number of parameters of $\hat{W}$, whereas SliceGPT use it for  reducing the feature dimension of $X$.
>
> **Q3:**  In paragraph of “Fast OND”, authors claim that Figure 2 shows that the low-rank structure of the product $W_v W_o^T$ can be determined by one of the $W_v$ and $W_o$. But Figure 2 only shows that the trend of singular value in $W_v$ and $W_o$ but doesn’t show that they have similar singular vectors. So the motivation of performing SVD on one of $W_v$ and $W_o$ seems not very solid.
>
> **A3:** Thanks for your insightful comments.   Since $rank(W_v W_o^T)< min(rank(W_v), rank(W_o))$, the low-rank property of $W_v W_o^T$ is upper bounded by the ranks of $W_v$ and $W_o$. If either $W_v$ or $W_o$ exhibits the low-rank structure, we immediately know that $W_v W_o^T$ is also a low-rank matrix. As a result, we only need to examine the distribution of the singular values of $W_v$ and $W_o$ to determine their low-rank structures. We will state these more clearly in the final manuscript.

---

> > ### Comment · Reviewer_ZyT3 · 2025-04-02
> >
> > >> Thanks for your insightful comments. Since $rank(W_v W_o^T)< min(rank(W_v), rank(W_o))$, the low-rank property of $W_v W_o^T$ is upper bounded by the ranks of $W_v$ and $W_o$. If either $W_v$ or $W_o$ exhibits the low-rank structure, we immediately know that $W_v W_o^T$ is also a low-rank matrix. As a result, we only need to examine the distribution of the singular values of $W_v$ and $W_o$ to determine their low-rank structures. We will state these more clearly in the final manuscript.
> >
> > My concern wasn't solved. It seems authors are trying to conflating the concept of "low-rank structure". In this reply, "A and B has similar low-rank structure"="A and B has similar rank", however, in section 3.2, "A and B has similar low-rank structure"="A and B has similar singular vectors". They are two different meanings.

---

> > > ### Author Response · Authors · 2025-04-03
> > >
> > > Dear Reviewer ZyT3:
> > >
> > >
> > > Many thanks for your time and efforts in reviewing our paper.
> > >
> > >
> > > **Q:** It seems authors are trying to conflating the concept of "low-rank structure". In this reply, “A and B has similar low-rank structure”=“A and B has similar rank”, however, in section 3.2, “A and B has similar low-rank structure"=“A and B has similar singular vectors". They are two different meanings.
> > >
> > >
> > > **A:** We sincerely appreciate your  comment on the “low-rank structure" and apologize for the confusion. Here,  the term  “low-rank structure" specifically refers to the low-rank property characterized by the singular values (i.e., the number of retained singular values required to preserve a certain energy ratio), rather than the singular vectors. Our focus on singular values, rather than singular vectors, stems from the fact that parameter redundancy is primarily determined by the proportion of truncated singular values. For instance, in cases where the singular values of a matrix exhibit rapid decay, a significant portion of the smaller singular values can be discarded while still maintaining high approximation accuracy. This rationale explains why, in Figure 2 of our paper, we exclusively presented the distribution of singular values. We will make changes in the final version of the manuscript as follows:
> > >
> > >
> > > **Old version:** Fortunately, we observe a symmetry property of  $W_v$ and $W_o$ (also for $W_q$ and $W_k$) as shown in Figure 2, which means that the low-rank structure  of the product $W_{v}W_o^{\top}$ can be  determined by one of $W_{v}$ and $W_{o}$. Therefore, we can only perform SVD on one of $W_v$ and $W_o$.
> > >
> > >
> > >
> > > **Revised version:** Fortunately, we observe  similar distributions of singular values  for  $W_v$ and $W_o$ (also for $W_q$ and $W_k$) as shown in Figure 2. This means that the number of retained singular values required to preserve a certain energy ratio for $W_v$ and $W_o$ are also similar. Therefore, we can only perform SVD on one of $W_v$ and $W_o$ to roughly determine the redundancy of the unified entity $W_{vo}$.

---

### Official Review · Reviewer_5bY6 · 2025-03-17

**Overall Recommendation:** 3

**Summary:**

This paper proposed a new structured pruning method by treating the matrix products WqWk and also WvWo as unified entities and applying PCA, and pruning the unimportant information. They also pruned the FFN layer and introduced a linear calibration method to reconstruct the residual error with two low-rank matrices. The experimental results show that the method achieved a SOTA result.

## update after rebuttal
The author addresses my concerns. I will keep my score as positive.

**Claims And Evidence:**

The claims made in the submission are supported by clear and convincing evidence.

**Essential References Not Discussed:**

No

**Experimental Designs Or Analyses:**

I check the soundness and validity of experimental designs and analyses, and they are correct.

**Methods And Evaluation Criteria:**

The proposed methods and evaluation criteria make sense for the problem or application at hand.

**Other Comments Or Suggestions:**

See weaknesses above.

**Other Strengths And Weaknesses:**

Strengths:

1. This paper is well-written and technically sound.

2. The way of using PCA to directly prune the output of WqWq and WvWo is somewhat novel.

3. The experimental results show that the proposed method performs well.

Weaknesses:

1. The product of $W_{q_i}W_{k_i}^\top$ has already shown the low-rank property since $d_h$ is smaller than $d$. Based on this, are you trying to reduce $d_h$ into an even smaller $r$ using PCA?

2. I am confused about Eq.7. Is $f(X)=XW$ in Eq.7? If so, this means that your target is to learn $\hat f(X)=0$, which is $\hat W=0$. If not, what are you trying to learn in Eq.7? The authors should further explain this.

3. There is a lack of some ablation studies. For example, adjusting the pruning ratio of FFN layers and MHA layers,

4. There is a lack of comparison to other SOTA methods, such as OSSCAR [1] and FLAP [2].

5. Typo error: at the end of page 1, the the -> the.

[1] OSSCAR: One-Shot Structured Pruning in Vision and Language Models with Combinatorial Optimization.

[2] Fluctuation-based Adaptive Structured Pruning for Large Language Models.

**Questions For Authors:**

See weaknesses above.

**Relation To Broader Scientific Literature:**

This paper is related to structure pruning and LoRA.

**Theoretical Claims:**

There is no proof or theoretical claim.

---

> ### Author Rebuttal · Authors · 2025-04-01
>
> Many thanks for your time and efforts in reviewing our paper. We will fully address your concerns below.
>
> **Q1:** The product of $W_{q_i}W_{k_i}^{\top}$  has already shown the low-rank property since $d_h$ is smaller than $d$. Based on this, are you trying to reduce $d_h$ into an even smaller $r$ using PCA ?
>
> **A1:**  Thanks for your question. Indeed, our orthogonal neuron decomposition (OND) method is to reduce the dimensionality of each attention head, i.e, reduce $d_h$  into a smaller $r$ using PCA.
>
> **Q2:** I am confused about Eq.7. Is $f(X)=XW$ in Eq.7? If so, this means that your target is to learn  $\hat{f}(X)=0$, which is $\hat{W}=0$. If not, what are you trying to learn in Eq.7? The authors should further explain this.
>
> **A2:** Sorry for the confusion. In Eq.7, the $f(X)$ is denoted as the original FFN layer. Our objective  is to recover the residual errors $E$ of pruned layers $\hat{f}$, that is, $E=f(X)-\hat{f}(X)$, where $f$ is the original version of $\hat{f}$. We approximate $E$ by a linear model: $E\approx X\hat{W}$, and thus the pruned layer $\hat{f}$ can be calibrated to approximate its original version: $f(X) \approx \hat{f}(X) + X\hat{W}$. The $\hat{f}$ is obtained by the algorithm detailed in the **FFN Pruning** section of our paper and both $f(X)$ and $\hat{f}(X)$ are  fixed in Eq.7. So the objective of Eq.7 is to learn the parameters $\hat{W}$ of the liner model  given both $f$ and $\hat{f}$ fixed.
>
> **Q3:** There is a lack of some ablation studies. For example, adjusting the pruning ratio of FFN layers and MHA layers.
>
> **Table1:** Ablation study of different pruning ratios of FFN layers and MHA layers (LLaMA-7B).
> | MHA | FNN | PPL ($\downarrow$) |Avg. Acc. ($\uparrow$) |
> |-------|-------|-------|-------|
> | 20% | 20% | 15.35 |64.54 |
> | 30% | 20% | 16.68 |63.21 |
> | 20% | 30% | 19.11 |61.57 |
> | 40% | 20% | 20.85|60.82 |
> | 20% | 40% | 24.21 |58.07 |
>
> **A3**:  Thanks for this valuable suggestion. We conducted ablation studies with different pruning ratios of FFN layers and MHA layers in the Table 1. We observe that the MHA layers can tolerate a larger pruning ratio than the FFN layers, meaning that the core design of Transformers, i.e., the multi-head attention layer, is particularly redundant. This observation coincides  with existing works such as [1].
>
> [1] What matters in transformers? not all attention is needed. https://arxiv.org/abs/2406.15786.
>
> **Q4:**  There is a lack of comparison to other SOTA methods, such as OSSCAR and FLAP.
>
> **Table2:** Comparisons with FLAP (Accuracy  ($\uparrow$)).
> |Method | PPL | BoolQ  | PIQA| HellaS |WinoG| ARC-e|ARC-c| OBQA |Avg.|
> |-------|-------|-------|-------|-------|-------|-------|-------|-------|-------|
> |FLAP | 17.0 | 69.4| 74.7| 66.9| 66.3| 64.6| 36.5 |38.2|59.5|
> | Olica (Ours) |**15.4**  | **71.6** | **77.9** |**77.3** |**70.0** |**72.1** |**42.7** |**44.2** |**64.5** |
>
> **Table3:** Comparisons with OSSCAR (PPL ($\downarrow$) on WikiText).
> | Method | OPT-1.3B | OPT-2.7B |OPT-6.7B |
> |-------|-------|-------|-------|
> | OSSCAR | **15.38** | 13.17 | 12.79 |
> | Olica (Ours) | 16.14 | **12.78** |**11.63** |
>
> **A4:** We compare Olica with FLAP  and OSSCAR in Table 2 and Table 3 in terms of accuracy and PPL, respectively. We can see that the proposed Olica consistently outperforms FLAP with large margins. Besides, our approach surpasses OSSCAR on the larger models, i.e., OPT-2.7B and OPT-6.7B.

---

### Decision · Program_Chairs · 2025-05-01

**Decision:**

Accept (poster)

**Comment:**

This manuscript proposes a methodology for compressing transformer architectures based on SVD compression of certain matrices. To correct for errors introduced during the compression a calibration step is used. The method is efficient and performs well on the, somewhat limited, set of evaluations. Nevertheless, I am sensitive to the issues of requiring/requesting excessive experiments given the breadth of work in the area. Moreover, the rebuttal phase did partially alleviate some of these concerns.

The reviewers do not identify major shortcomings (rather the manuscript will simply get stronger with more experiments), and the primary strength of the manuscript is as a well executed empirical demonstration of a simple methodology that leverages common tools. Therefore, my inclination is positive—there are good ideas here that may form the basis for future study and incorporation into other methods.

Assorted notes:

The section on MHA pruning conflates eigenvectors and singular vectors. This needs to be corrected.

As the reviewers have pointed out, the fast OND part needs some clarification. The suggestions below address the concerns, but as written the procedure could certainly over-estimate the numerical rank.

Per the reviewers comments, some of the presentation can be improved and it should be.